# BETTER BOUNDS FOR THE DISTRIBUTED EXPERTS PROBLEM

**David P. Woodruff**
Carnegie Mellon University
dwoodruf@andrew.cmu.edu

**Samson Zhou**
Texas A&M University
samsonzhou@gmail.com

## ABSTRACT

In this paper, we study the distributed experts problem, where $n$ experts are distributed across $s$ servers for $T$ timesteps. The loss of each expert at each time $t$ is the $\ell_p$ norm of the vector that consists of the losses of the expert at each of the $s$ servers at time $t$. The goal is to minimize the regret $R$, i.e., the loss of the distributed protocol compared to the loss of the best expert, amortized over the all $T$ times, while using the minimum amount of communication. We give a protocol that achieves regret roughly $R \gtrsim \frac{1}{\sqrt{T} \cdot \operatorname{poly} \log(nsT)}$, using $\mathcal{O}\left(\frac{n}{R^2} + \frac{s}{R^2}\right) \cdot \max(s^{1-2/p}, 1) \cdot \operatorname{poly} \log(nsT)$ bits of communication, which improves on previous work.

## 1 INTRODUCTION

Many modern applications require sequential decisions based on predictions from a large set of distributed experts. For example, in hyperparameter optimization or model selection, each expert corresponds to different architectures or pre-trained models evaluated across separate datasets, and the algorithm must dynamically choose which to deploy. In recommendation systems, experts encode distinct ranking functions, with feedback only from selected actions, while in reinforcement learning, experts correspond to policies trained on separate environments. Across these applications, the decision-maker must aggregate information from multiple sources and adapt to evolving losses in distributed systems. This setting is formalized by the online prediction with expert advice problem, where $n$ experts incur losses at each step over $T$ rounds, and the algorithm selects an expert based on observed history to minimize cumulative regret, defined relative to the best expert in hindsight. Classical algorithms, including the Exponential Weights Algorithm (EWA) and Multiplicative Weights Update (MWU), achieve optimal regret $\mathcal{O}\left(\sqrt{\frac{\log n}{T}}\right)$ in the full-information setting (Arora et al., 2012), while EXP3 attains near-optimal regret $\mathcal{O}\left(\sqrt{\frac{n \log n}{T}}\right)$ in the bandit setting (Auer et al., 2002). As the number $n$ of experts and prediction rounds $T$ grow, running these algorithms centrally becomes computationally expensive, motivating the study of scalable models. Prior work has explored streaming approaches (Srinivas et al., 2022; Peng & Zhang, 2023; Woodruff et al., 2023; Peng & Rubinstein, 2023; Aamand et al., 2023; Braverman et al., 2026), which process expert losses sequentially while maintaining only compact summaries of historical data.

In this work, we consider a complementary challenge: *distributed online learning with experts*. Here, expert costs are partitioned across $s$ servers, and a central coordinator aims to implement a low-regret algorithm while minimizing communication with the servers. This distributed formulation arises naturally in large-scale online optimization and hyperparameter tuning across multiple datasets, or in settings where each server holds a private subset of data. For example, each expert may correspond to a model trained on the aggregated data across servers, with the total loss computed as a sum of per-server losses, as in the HPO-B benchmark (Pineda-Arango et al., 2021). Other aggregation functions, such as the maximum loss across servers, can also be relevant when individual server costs are constrained, highlighting the flexibility of the framework to accommodate different practical objectives.

Our focus is on understanding the tradeoff between communication and regret in distributed online learning with experts. Unlike memory-constrained streaming approaches, the coordinator can maintain full information on all experts, so the primary challenge is minimizing communication. We study this in the message-passing model, where the coordinator communicates privately with each server in sequential rounds. Prior work has largely focused on $\ell_1$ loss (Jia et al., 2025), but many applications—such as risk-sensitive optimization, robust model selection, and distributed control—require general $\ell_p$ losses with $p > 1$. Different values of $p$ balance robustness and penalization of large deviations: $\ell_\infty$ minimizes the maximum loss, intermediate values like $\ell_2$ moderately penalize large deviations, and values near $\ell_1$ emphasize outlier robustness. For example, $\ell_2$ loss is extensively studied in streaming settings due to connections with entropy (Harvey et al., 2008; Woodruff & Zhou, 2021). We therefore develop algorithms that achieve near-optimal regret for general $\ell_p$ losses while keeping communication provably low.

## 1.1 Distributed Online Learning with Experts in the Coordinator Model

We consider the distributed online learning with experts problem in the coordinator (message-passing) model. There are $s$ servers, denoted $[s] = \{1, 2, \ldots, s\}$. At each round $t \in [T]$, each server $j \in [s]$ receives a local loss vector $(\ell_1(j, t), \ldots, \ell_n(j, t)) \in \mathbb{R}^n$. The goal is to minimize the cumulative regret of the algorithm while using minimal communication between servers.

**Online learning with experts.** In the online learning with experts problem, there are $n$ experts who make predictions across a time horizon of length $T$. For each time $t \in [T]$, an algorithm selects an expert $i_t \in [n]$ and then observes the loss vector $(L_1(t), \ldots, L_n(t))$, incurring loss $L_{i_t}(t)$. The total loss of the algorithm is $\sum_{t \in [T]} L_{i_t}(t)$, and the best expert $i^*$ incurs $\sum_{t \in [T]} L_i(t)$. The regret of the algorithm is defined by

$$R = \frac{1}{T} \left( \sum_{t \in [T]} L_{i_t}(t) - \sum_{t \in [T]} L_{i^*}(t) \right),$$

which the algorithm aims to minimize. In the distributed setting, the losses $L_i(t)$ are not given explicitly. Instead, the loss of each is computed across servers via the $L_p$ loss $L_i(t) = \left( \sum_{j \in [s]} \ell_i(j, t)^p \right)^{1/p}$, where $p \geq 1$ is fixed and known in advance. This distributed formulation captures scenarios where expert evaluations are partitioned across multiple servers or datasets.

**Communication complexity.** Since the losses $L_i(t)$ are not explicitly given, the servers must collectively execute a distributed protocol $\Pi$. Each server has private randomness, and all share public randomness. In the coordinator model, servers communicate only with the coordinator; in the message-passing model, direct server-to-server communication provides at most a constant-factor improvement, so we treat the two interchangeably. We assume $\Pi$ is sequential and round-based: in each round, the coordinator interacts with a subset of servers and waits for responses. The protocol is self-delimiting so servers know when a round concludes. Let $\Pi(r)$ denote the transcript of round $r$, with $|\Pi(r)|$ its communication cost. The total communication is $\sum_r |\Pi(r)|$. Designing protocols that balance total communication with low regret is the central challenge in distributed online learning with experts.

## 1.2 Our Contributions

In this paper, we study the distributed online learning with experts problem. As a warm-up, we show that there exists a distributed protocol that achieves a nearly optimal regret of $\mathcal{O}\left( s^{1/p} \sqrt{\frac{\log n}{T}} \right)$, with communication $\tilde{\mathcal{O}}\left( nT + sT \right)$:

**Theorem 1.1.** *Let $b > a > 0$ be fixed constants and suppose $\ell_i(j, t) \in [a, b]$ for all $t \in [T]$, $i \in [n]$ and $j \in [s]$. There exists an algorithm that achieves expected regret at most $\mathcal{O}\left( s^{1/p} \sqrt{\frac{\log n}{T}} \right)$ and with high probability, uses total communication at most $\mathcal{O}\left( sT \right) + nT \cdot \mathrm{polylog}(nsT)$ bits.*

By comparison, the work of Jia et al. (2025) only achieved distributed protocols for general $\ell_p$ losses for the broadcast/blackboard model, where servers can publicly broadcast information that

is visible to other servers and the communication cost of a protocol is then the total length of the broadcast messages. This setting is "easier" than the message-passing model since sending the same message to all servers in our setting would incur $s$ communication multiplicative overhead. Thus the techniques are fundamentally different. For instance, the $\ell_p$ loss algorithm by Jia et al. (2025) for the broadcast model walks through all servers to find the server $j \in [s]$ with the largest loss for each expert $i \in [n]$. While each server $j$ only needs to send values larger than the previously broadcast losses for each expert in the broadcast model, resulting in roughly total communication $\tilde{\mathcal{O}}(s + n)$, such an approach would require $\tilde{\mathcal{O}}(ns)$ communication in our setting.

On the other hand, Jia et al. (2025) achieved a distributed protocol for the SUM problem in the message-passing model, i.e., $\ell_1$ loss, with $\tilde{\mathcal{O}}(nT) + \mathcal{O}(Ts)$ total communication, for regret $\mathcal{O}\left(s^{1/p}\sqrt{\frac{\log n}{T}}\right)$. However, because $\ell_1$ loss is additive across servers, then standard techniques such as sampling a number of losses to be communicated, with probability proportional to their magnitudes, can work for $\ell_1$ loss but fail to work for sub-additive or super-additive losses such as $\ell_p$ losses. In contrast, Theorem 1.1 is able to achieve optimal regret across all $\ell_p$ losses.

We remark that Jia et al. (2025) assumed the losses are normalized so that the total loss for a single expert on a single time is at most 1. By comparison, if each server is permitted to have loss $[a, b]$ by our assumption, then all experts have loss $\Omega(s^{1/p})$, which accounts for the difference in their stated bounds and our referenced bounds. We can parameterize Theorem 1.1 to obtain a more general communication-regret trade-off as follows:

**Theorem 1.2.** *Let $b > a > 0$ be fixed constants and suppose $\ell_i(j, t) \in [a, b]$ for all $t \in [T]$, $i \in [n]$ and $j \in [s]$. Let $R \geq \frac{1}{\sqrt{T}}$. Then there exists an algorithm that achieves expected regret at most $\mathcal{O}\left(Rs^{1/p}\sqrt{\log n}\right)$ and with high probability, uses total communication at most $\left(\frac{n+s}{R^2}\right) \cdot$ polylog$(nsT)$ bits.*

We emphasize that seeking regret $R \geq \frac{1}{\sqrt{T}}$ is standard, because it is information-theoretically impossible to achieve regret $R < \frac{1}{\sqrt{T}}$ (Cover, 1966).

In contrast to Theorem 1.2, Jia et al. (2025) achieved a distributed protocol for the SUM problem, i.e., $\ell_1$ loss, with $\tilde{\mathcal{O}}\left(\frac{n}{R^2}\right) + \mathcal{O}(Ts)$ total communication, for regret $R$. Thus not only does Theorem 1.2 improve on the result of Jia et al. (2025) for general $\ell_p$ losses, but it also improves upon the $\mathcal{O}(Ts)$ dependency in the result of Jia et al. (2025) to parameterization across general regret $R$. For example, with regret $R = \mathcal{O}(1)$, the protocol of Jia et al. (2025) still has dependency $\mathcal{O}(Ts)$ while our protocol has dependency $\mathcal{O}(s)$, which is substantially less for large time horizons $T$. We summarize this discussion in Figure 1.

| Reference | Loss function | Communication cost (bits) |
|---|---|---|
| Jia et al. (2025) | $\ell_1$ loss | $\left(\frac{n}{R^2} + Ts\right) \cdot$ polylog$(nsT)$ |
| Theorem 1.2 | $\ell_p$ loss | $\left(\frac{n+s}{R^2}\right) \cdot$ polylog$(nsT)$ |

Fig. 1: Our work is the first to study $\ell_p$ loss in the coordinator model; for the special case of $p = 1$, we obtain better regret-communication tradeoffs for regret $R$.

We enable the handling of $\ell_p$ losses is to embed $\ell_p$ into $\ell_\infty$ through the use of exponential random variables, similar to the blackboard protocol of Jia et al. (2025). However, the servers cannot easily find the maximum in the coordinator model, so we require a careful analysis of subsampling and thresholding to ensure that we find the maximum without too many servers sending low values. Additionally, the variance of the resulting estimator is unbounded, so we utilize a geometric mean estimator. These novelties form the basis of our main algorithmic contribution. Although similar approaches have been used in other contexts such as norm estimation in the streaming model (Li, 2008; Woodruff & Zhou, 2021), this is the first time these techniques have been applied to online learning in the distributed setting, to the best of our knowledge. As a result, we can handle $\ell_p$ losses, which previous works cannot handle (Jia et al., 2025). We provide more details in Section 3 and Section 4. Finally, we remove the assumption that the losses are between a range of constants $[a, b]$.

**Theorem 1.3.** *Suppose we have $\ell_i(j, t) \leq 1$ for all $t \in [T]$. There exists an algorithm that achieves expected regret at most $\mathcal{O}\left(Rs^{1/p}\sqrt{\log n}\right)$ and with high probability, uses total communication at most $\left(\frac{n+s}{R^2}\right) \cdot \max(s^{1-2/p}, 1) \cdot \operatorname{polylog}(nsT)$ bits.*

Finally, to complement our theoretical results, we conduct a number of empirical evaluations as a simple proof-of-concept. These results appear in Section 6.

**Algorithmic and technical novelties.** The major challenge is that $\ell_p$ losses are significantly more nuanced than $\ell_1$ losses. Observe that since $\ell_1$ loss is additive, then the total loss is the sum of the individual losses. Thus, it is natural to sample individual losses with probability proportional to their magnitude. For $\ell_p$ losses, it may be possible to perform some more nuanced sampling, but it is not clear how to efficiently perform sampling in distributed models across all of the experts. To overcome this barrier, we embed $\ell_p$ losses into an $\ell_\infty$ framework using random exponential scalings, such that the largest scaled contribution approximates the original $\ell_p$ loss. However, due to the probability density function of exponential random variables, the resulting distribution has unbounded variance and thus we use a geometric mean of an independent number of random scalings to acquire an unbiased estimator with bounded variance. We then interface these estimated losses with a multiplicative weights update (MWU) algorithm for the purpose of online learning. Notably, the use of a geometric mean estimator to reduce variance of estimators using exponential random variables is both an algorithmic and a technical novelty for the distributed online learning with experts problem that we believe may have independent interest to other applications. In principle, the geometric mean estimator could be potentially replaced by other variance-reduction techniques; the design of such algorithms is an interesting direction for future work.

## 2 PRELIMINARIES

We briefly discuss a number of relevant preliminaries necessary for our algorithms and analysis. We first recall the following standard formulation of Chernoff bounds for concentration inequalities.

**Theorem 2.1** (Chernoff Bounds). *Let $X_1, X_2, \ldots, X_n$ be independent Bernoulli random variables with $\mathbf{Pr}\left[X_i = 1\right] = p_i$ and $\mathbf{Pr}\left[X_i = 0\right] = 1 - p_i$. Define $X = \sum_{i=1}^n X_i$ and let $\mu = \mathbb{E}[X] = \sum_{i=1}^n p_i$. Then for any $\delta > 0$, we have $\Pr\left(|X - \mu| \geq \delta\mu\right) \leq 2\exp\left(-\frac{\delta^2 \mu}{3}\right)$.*

We next define an exponential random variable.

**Definition 2.2** (Exponential random variable). *An **exponential random variable** with rate parameter $\lambda > 0$ is a continuous random variable $X$ with probability density function (PDF) given by:*

$$f_X(x) = \begin{cases} \lambda e^{-\lambda x}, & x \geq 0, \\ 0, & x < 0. \end{cases}$$

We show the following max stability property of exponential random variables.

**Lemma 2.3.** *[Max stability of exponential random variables] Let $X_i = \frac{f_i}{e_i^{1/p}}$ for $i \in [n]$, where $e_1, \ldots, e_n$ are independent exponential random variables with rate $1$. Then $\max_{i \in [n]} \frac{f_i}{e_i^{1/p}}$ is distributed as $\frac{\|f\|_p}{e^{1/p}}$, where $e$ is an exponential random variable with rate $1$.*

Finally, we recall the multiplicative weights update (MWU) algorithm. For $\eta$ roughly $\frac{1}{\sqrt{T}}$, the following guarantees are known on the MWU algorithm in Algorithm 1:

**Theorem 2.4.** *For a set of $n$ experts with the second moment of the loss bounded by at most $\rho$ on each of $T$ rounds, the multiplicative weights update (MWU) algorithm achieves expected regret $\mathcal{O}\left(\sqrt{\frac{\rho \log n}{T}}\right)$.*

## 3 WARM-UP: SIMPLE ALGORITHM

We present a simple distributed algorithm that achieves near-optimal regret, using at most $\mathcal{O}\left(sT\right) + nT \cdot \operatorname{polylog}(nsT)$ total bits of communication. The intuition for the algorithm is as follows.

---

**Algorithm 1** Multiplicative weights update algorithm

---

**Input:** Learning rate $\eta$, losses $\{\ell_i(t)\}$ for all times $t \in [T]$, experts $i \in [n]$
**Output:** Sequence of experts to play on each day
1: $w_i \leftarrow 0$ for all $i \in [n]$
2: **for** each time $t \in [T]$ **do**
3:     **for** each $i \in [n]$ **do**
4:         $w_i \leftarrow w_i + \ell_i(t)$
5:     **end for**
6:     Sample $i \in [n]$ with probability proportional to $\exp(-\eta w_i)$
7: **end for**

---

Suppose for all experts $i \in [n]$ and times $t \in [T]$, we have the loss of $i$ at time $t$, $L_i(t) = \left( \sum_{j \in [s]} \ell_i(j, t)^p \right)^{1/p}$. Then by setting $w_i(t) = \sum_{t' < t} L_i(t)$, we can run the standard Multiplicative Weights Update (MWU) framework with weights $w_i(t)$ by playing expert $i$ with probability proportional to $\exp(-\eta w_i(t))$ for a fixed learning rate $\eta > 0$. Unfortunately, we do not have the losses $\{L_i(t)\}$. Instead, for time $t \in [T]$, each expert $i \in [n]$, and each server $j \in [s]$, we generate an exponential random variable $e_i(j, t)$. By the max-stability property of exponential random variables, c.f., Lemma 2.3, we have

$$\max_{j \in [s]} \frac{\ell_i(j, t)}{(e_i(j, t))^{1/p}} \sim \frac{\left( \sum_{j \in [s]} \ell_i(j, t)^p \right)^{1/p}}{e^{1/p}},$$

where $e$ is another exponential random variable. Note that the right-hand side is exactly the $\ell_p$ loss of $i$ on time $t$. Moreover, we can compute the expected value of $\frac{1}{e^{1/p}}$, so that if the servers can simply send the maximum of the scaled losses $\frac{\ell_i(j,t)}{(e_i(j,t))^{1/p}}$, then we can compute an unbiased estimate to the $\ell_p$ loss $\left( \sum_{j \in [s]} \ell_i(j, t)^p \right)^{1/p}$.

Unfortunately, there are two challenges with this approach. First, the servers do not know how to send the maximum. Second, the variance of the resulting random variable is unbounded. To handle the first issue, we first show that with high probability, the maximum is above some threshold that is roughly $s^{1/p}$ and moreover, there is only a small number of scaled losses that are above this threshold. Hence, we can upper bound the total amount of communication. To address the second issue, we use a standard estimator (Li, 2008) that takes the geometric mean of a constant number of these estimators, which lowers the variance to a small amount. We again remark that this pipeline serves as our main algorithmic contribution over previous works such as Jia et al. (2025); the algorithm appears in full in Algorithm 2, where $\widehat{s_i(t)}$ is the geometric mean estimator for the loss $L_i(t)$ of expert $i$ at time $t$ and $w_i(t)$ is the resulting weight for expert $i$ input to MWU.

We first recall the following fact about probability density functions for polynomials of random variables.

**Fact 3.1.** *Suppose a random variable $X$ has probability density function $f(x)$, with support on the non-negative reals. Then the random variable $X^{-1/p}$ has probability density function $px^{-p-1} \cdot f(x^{-p})$.*

We now compute the expectation and upper bound the variance of our geometric mean estimator.

**Lemma 3.2.** *[Expectation and variance of geometric mean estimator] Let $p > 0$ be fixed and $R \geq 3p$ be an integer. Let $e_1, \ldots, e_B$ be independent exponential random variables with rate 1. Let $Z_b = \frac{1}{e_b^{1/p}}$ for all $b \in [B]$, and let $Z = \left( \prod_{b \in [B]} Z_b \right)^{1/B}$ be their geometric mean. Then there exists a universal constant $C_{3.2} \in (0, 2^B)$ such that $\mathbb{E}[Z] = C_{3.2}$ and $\mathbb{E}[Z^2] \leq 3^B$.*

We now show a structural property showing how estimates $\widehat{s_i(t)}$ to losses $L_i(t) = \left( \sum_{j \in [s]} \ell_i(j, t)^p \right)^{1/p}$ that are roughly unbiased and have small second moment facilitate an implementation of the MWU algorithm with bounded regret.

---

**Algorithm 2** Distributed protocol with near-optimal regret

---

**Input:** Losses $\{\ell_i(j,t)\}$ for all times $t \in [T]$, experts $i \in [n]$, servers $j \in [s]$, $L_p$ loss parameter $p$
**Output:** Sequence of experts to play on each day

1: $B \leftarrow \left\lceil \frac{3}{p} \right\rceil$
2: **for** each time $t \in [T]$ **do**
3:    **for** each $i \in [n]$ **do**
4:       **for** each server $j \in [s]$ **do**
5:          Let $\ell_i(j,t)$ be the loss of expert $i$ on server $j$ at time $t$
6:          Let $e_i^{(b)}(j,t)$ be independent exponential random variables for all $b \in [B]$
7:          $q_i^{(b)}(j,t) \leftarrow \frac{\ell_i(j,t)}{C_{3.2}(e_i^{(b)}(j,t))^{1/p}}$ for $b \in [B]$         ▷Lemma 3.2
8:          **if** $q_i^{(b)}(j,t) \geq \frac{s^{1/p}}{100\log(nsT)}$ **then**
9:             Send $q_i^{(b)}(j,t)$ to coordinator
10:          **end if**
11:       **end for**
12:    **end for**
13:    **for** each $i \in [n]$ **do**
14:       $\widehat{s_i(t)} \leftarrow \prod_{b \in [B]} \left( \max_{j \in [s]} q_i^{(b)}(j,t) \right)^{1/B}$
15:       $w_i(t) \leftarrow w_i(t-1) + \widehat{s_i(t)}$
16:    **end for**
17:    Play MWU on $\{w_i(t)\}$
18: **end for**

---

**Lemma 3.3.** *Suppose $\mathbb{E}\left[\widehat{s_i(t)}\right] = L_i(t)$ and this random variable has second moment at most $\rho$ for all $i \in [n]$ and $t \in [T]$. Suppose the multiplicative weights update (MWU) algorithm is executed with loss sequences $\widehat{s_i(t)}$ instead of $L_i(t)$ for all $i \in [n]$ and $t \in [T]$. Then the sequence of resulting choices achieves expected regret $\mathcal{O}\left(\sqrt{\frac{\rho\log n}{T}}\right)$ on the sequence of losses $\{L_i(t)\}_{i \in [n], t \in [T]}$.*

Unfortunately, we cannot immediately apply Lemma 3.3 because we do not have $\mathbb{E}\left[\widehat{s_i(t)}\right] = L_i(t)$. Namely, in the case where $\max_{j \in [s]} q_i^{(b)}(j,t) < \frac{s^{1/p}}{100\log(nsT)}$, then the value is not sent to the coordinator. Fortunately, we show this only holds with probability $\frac{1}{\text{poly}(nT)}$ due to the distribution of exponential random variables. As a result, the overall regret is only changed by lower-order terms.

Without loss of generality, we set the fixed constants $b > a > 0$ to be $[1,5]$; our proofs easily generalize to intervals $[a, b]$. Using standard concentration inequalities, e.g., Theorem 2.1, we now upper bound the expected regret of our simple algorithm and the total communication of our warm-up distributed protocol.

**Lemma 3.4.** *Suppose $\ell_i(j,t) \in [1,5]$ for all $t \in [T]$, $i \in [n]$ and $j \in [s]$. Then the expected regret of Algorithm 2 is at most $\mathcal{O}\left(s^{1/p}\sqrt{\frac{\log n}{T}}\right)$.*

**Lemma 3.5.** *Suppose $\ell_i(j,t) \in [1,5]$ for all $t \in [T]$, $i \in [n]$ and $j \in [s]$. Then with high probability, the total communication of Algorithm 2 is at most $\mathcal{O}(sT) + nT \cdot \text{polylog}(nsT)$ bits.*

Combining Lemma 3.4 and Lemma 3.5, we have Theorem 1.1.

## 4 COMMUNICATION-REGRET TRADE-OFF

In this section, we parameterize our warm-up algorithm to achieve a communication-regret trade-off. Namely, we show that if the goal is to achieve regret $R$, then there exists a distributed protocol that uses total communication at most $\left(\frac{n+s}{R^2}\right) \cdot \text{polylog}(nsT)$ bits.

The algorithm is quite similar to Algorithm 2. The main difference is that now at each time, each server is sampled to run the previous protocol. That is, with probability $\varrho$, each server independently performs the same protocol as before. Otherwise, with probability $1 - \varrho$, the server does not speak. We remark that the coordinator can also know the outcome of these events by using public randomness, so that the coordinator does not need to speak to each server to determine whether the server was sampled. The algorithm appears in full in Algorithm 3.

---

**Algorithm 3** Distributed protocol with communication-regret trade-off

---

**Input:** Target regret $R$, losses $\{\ell_i(j,t)\}$ for all times $t \in [T]$, experts $i \in [n]$, and servers $j \in [s]$, $L_p$ loss parameter $p$

**Output:** Sequence of experts to play on each day

1: $B \leftarrow \left\lceil \frac{3}{p} \right\rceil, \varrho \leftarrow \frac{1}{R^2 T}$
2: **for** each time $t \in [T]$ **do**
3:      With probability $1 - \varrho$, the servers do not send anything and the weights $\{w_i(t)\}$ are not updated
4:      Otherwise, with probability $\varrho$, do the following:
5:      **for** each $i \in [n]$ **do**
6:         **for** each server $j \in [s]$ **do**
7:            Let $\ell_i(j,t)$ be the loss of expert $i$ on server $j$ at time $t$
8:            Let $e_i^{(b)}(j,t)$ be independent exponential random variables for all $b \in [B]$
9:            $q_i^{(b)}(j,t) \leftarrow \frac{\ell_i(j,t)}{C_{3.2}(e_i^{(b)}(j,t))^{1/p}}$ for $b \in [B]$        ▷Lemma 3.2
10:            **if** $q_i^{(b)}(j,t) \geq \frac{s^{1/p}}{100 \log(nsT)}$ **then**
11:                Send $q_i^{(b)}(j,t)$ to coordinator
12:            **end if**
13:         **end for**
14:      **end for**
15:      **for** each $i \in [n]$ **do**
16:         $\widehat{s_i(t)} \leftarrow \frac{1}{\varrho} \cdot \prod_{b \in [B]} \left( \max_{j \in [s]} q_i^{(b)}(j,t) \right)^{1/B}$
17:         $w_i(t) \leftarrow w_i(t-1) + \widehat{s_i(t)}$
18:      **end for**
19:      Play MWU on $\{w_i(t)\}$
20: **end for**

---

We first show the following property of exponential random variables.

**Fact 4.1.** *Let $e$ be an exponential random variable with rate $1$. Then for all $x > 2$, we have* $\mathbf{Pr}\left[ \frac{1}{e} \in (x, 2x] \right] \in \left[ \frac{1}{4x}, \frac{1}{2x} \right]$.

We now analyze the expectation and upper bound the variance of our communication-regret trade-off protocol, using the properties of exponential random variables.

**Lemma 4.2.** *We have* $\mathbb{E}\left[ \widehat{s_i(t)} \right] = L_i(t) + \frac{1}{\text{poly}(nT)}$ *and* $\mathbb{E}\left[ (\widehat{s_i(t)})^2 \right] \leq \mathcal{O}\left( TR^2 \right) \cdot 3^B \cdot (L_i(t))^2$.

Again, we remark that we do not have the idealized expectation of $\mathbb{E}\left[ \widehat{s_i(t)} \right] = L_i(t)$ because $\max_{j \in [s]} q_i^{(b)}(j,t)$ may not be sent to the coordinator if it is less than $\frac{s^{1/p}}{100 \log(nsT)}$, which changes the resulting expectation calculation. However, since this event only occurs with probability $\frac{1}{\text{poly}(nT)}$, then the resulting claim follows. We then upper bound the regret and the total communication of Algorithm 3 as follows:

**Lemma 4.3.** *Suppose $\ell_i(j,t) \in [1,5]$ for all $t \in [T]$, $i \in [n]$ and $j \in [s]$. The expected regret of Algorithm 3 is at most $\mathcal{O}\left( Rs^{1/p}\sqrt{\log n} \right)$.*

**Lemma 4.4.** *Suppose $\ell_i(j,t) \in [1,5]$ for all $t \in [T]$, $i \in [n]$ and $j \in [s]$. Then with high probability, the total communication is at most $\left( \frac{n+s}{R^2} \right) \cdot \text{polylog}(nsT)$ bits.*

Putting together Lemma 4.3 and Lemma 4.4, we have Theorem 1.2.

## 5 FULL ALGORITHM

In this section, we build on our previous algorithms to present our main result, which achieves regret roughly $R$ while using $\left(\frac{n+s}{R^2}\right) \cdot \max(s^{1-2/p}, 1) \cdot \mathrm{polylog}(nsT)$ total bits of communication. Unlike the previous sections, we do not require each loss to be within an interval $[a, b]$ where $b > a > 0$ are fixed constants. Algorithm 4 differs from previous algorithms in that it permits the servers to communicate more aggressively by sending smaller values, in particular the values larger than a smaller threshold. However, this can only happen when a server is appropriately sampled.

---

**Algorithm 4** Distributed protocol for the experts problem

---

**Input:** Target regret $R$, losses $\{\ell_i(j, t)\}$ for all times $t \in [T]$, experts $i \in [n]$, and servers $j \in [s]$, $L_p$ loss parameter $p$

**Output:** Sequence of experts to play on each day

1: $B \leftarrow \left\lceil \frac{3}{p} \right\rceil$, $\varrho \leftarrow \frac{1}{R^2 T} \cdot \max(s^{1-2/p}, 1)$, $A \leftarrow \lceil 10 \log(nsT) \rceil$

2: **for** each time $t \in [T]$ **do**

3:     With probability $1 - \varrho$, the servers do not send anything and the weights $\{w_i(t)\}$ are not updated

4:     Otherwise, pick an integer $a > 0$ with probability $\frac{\varrho}{2^a}$

5:     All servers send values $q_i^{(b)}(j, t)$ that are at least $\frac{s^{1/p}}{100 \cdot (2^a)^{1/p} \log(nsT)}$ to the coordinator

    ▷Lemma 3.2

6:     Let $\widehat{q_i^{(b)}}(j, t) = \frac{q_i^{(b)}(j,t)}{C_{3.2}}$ if received by the coordinator, or set to zero otherwise

7:     **for** each $i \in [n]$ **do**

8:         $\widehat{s_i(t)} \leftarrow \cdot \prod_{b \in [B]} \left( \max_{j \in [s]} \widehat{q_i^{(b)}}(j, t) \right)^{1/B}$

9:         Let $a^* \geq 0$ be the smallest non-negative integer such that $\widehat{s_i(t)} \geq \left( \frac{s}{2^{a^*}} \right)^{1/p}$

10:         **if** $\widehat{s_i(t)} \neq 0$ and $a^* \leq a$ **then**

11:             $w_i(t) \leftarrow w_i(t-1) + 2^{a^*} R^2 T \cdot \widehat{s_i(t)}$

12:         **end if**

13:     **end for**

14:     Play MWU on $\{w_i(t)\}$

15: **end for**

---

We first consider the expectation and variance of Algorithm 4. We use properties of exponential random variables to show that up to a small perturbation, the expected valued of $\widehat{s_i(t)}$ is precisely the loss $L_i(t)$ of expert $i$ on time $t$. We use a similar argument for the second moment of $\widehat{s_i(t)}$.

**Lemma 5.1.** *Suppose $q_i^{(b)}(j, t) \geq \frac{1}{100\sqrt{T}}$ for all $b \in [B]$. Then we have* $\mathbb{E}\left[ \widehat{s_i(t)} \right] = L_i(t) + \frac{1}{\mathrm{poly}(nT)}$ *and* $\mathbb{E}\left[ (\widehat{s_i(t)})^2 \right] \leq s^{2/p} \cdot R^2 T$.

Next, we upper bound the regret and total communication of Algorithm 4 as follows:

**Lemma 5.2.** *Suppose we have $\ell_i(j, t) \leq 1$ for all $t \in [T]$. The expected regret of Algorithm 4 is at most $\mathcal{O}\left( Rs^{1/p} \sqrt{\log n} \right)$.*

**Lemma 5.3.** *Suppose we have $\ell_i(j, t) \leq 1$ for all $t \in [T]$. Then with high probability, the total communication for Algorithm 4 is at most $\left( \frac{n+s}{R^2} \right) \cdot \max(s^{1-2/p}, 1) \cdot \mathrm{polylog}(nsT)$ bits.*

Putting together Lemma 5.2 and Lemma 5.3, we have our main result:

**Theorem 1.3.** *Suppose we have $\ell_i(j, t) \leq 1$ for all $t \in [T]$. There exists an algorithm that achieves expected regret at most $\mathcal{O}\left( Rs^{1/p} \sqrt{\log n} \right)$ and with high probability, uses total communication at most $\left( \frac{n+s}{R^2} \right) \cdot \max(s^{1-2/p}, 1) \cdot \mathrm{polylog}(nsT)$ bits.*

## 6 EMPIRICAL EVALUATIONS

In this section, we perform experimental evaluations on our distributed protocol, comparing it to both the standard offline MWU algorithm and the previous protocol of Jia et al. (2025). Specifically, we study the HPO-B dataset (Pineda-Arango et al., 2021), which is a benchmark designed to evaluate and compare the performance of hyperparameter optimization (HPO) algorithms across various machine learning tasks. We conducted our empirical evaluations using Python 3.11.5 on a 64-bit operating system with an Intel(R) Core(TM) i7-3770 processor, featuring 4 cores, a base clock speed of 3.4GHz, and 16GB of RAM. The dataset includes a diverse set of datasets and learning tasks, such as classification and regression, to assess the effectiveness, efficiency, and computational cost of different HPO methods. To support transparency and facilitate reproducibility of our results, we provide a complete implementation of all algorithms, experimental setups, and data pre-processing routines, which are available at https://github.com/samsonzhou/distributed-experts.

As a black-box benchmark for hyperparameter optimization, we can treat the various models in the HPO-B benchmark as distinct experts in the distributed experts problem, with different datasets assigned to different servers. Each search step, which is random search for all model classes, can be viewed as one day in the distributed experts scenario. The cost vector represents the normalized negative accuracy of the models on the different datasets during a search step. We thus compared our simple distributed protocol from Algorithm 2 against MWU and against the algorithm of Jia et al. (2025). Our results in Figure 2a show that for $p > 1$, the total communication increases as the threshold increases; however, the trend is reversed for $p < 1$. Figure 2b shows that our algorithm surprisingly has better reward than the MWU algorithm; we believe the latter is an issue of fine-tuning the learning rate of MWU. Finally, Figure 2c shows that for the special case of $p = 1$, our algorithm achieves better communication than the algorithm of Jia et al. (2025). Additionally, Figure 2c exhibits the communication-regret trade-off as expected from our theoretical analysis.

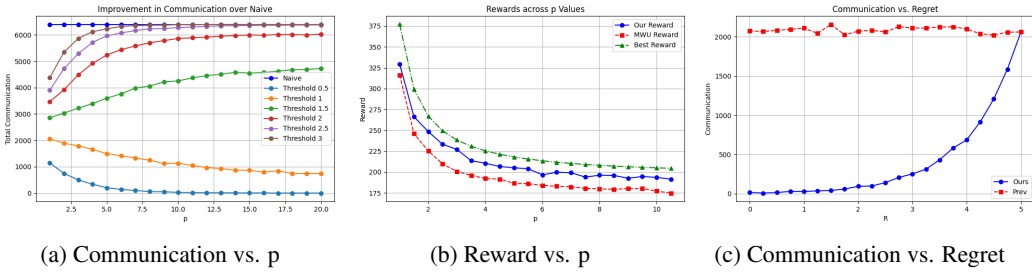

(a) Communication vs. p          (b) Reward vs. p          (c) Communication vs. Regret

**Conclusion.** In this paper, we studied the distributed online learning with experts problem in the coordinator model, where each expert's loss is implicitly defined across $s$ servers. We introduced the first protocol for general $\ell_p$ losses, achieving regret $R \gtrsim \frac{1}{\sqrt{T} \cdot \text{poly} \log(nsT)}$ with communication $\mathcal{O}\left(\left(\frac{n}{R^2} + \frac{s}{R^2}\right) \cdot \max(s^{1-2/p}, 1) \cdot \text{poly} \log(nsT)\right)$. For the important special case where nonzero server losses are bounded, our protocol attains regret $\mathcal{O}\left(Rs^{1/p}\sqrt{\log n}\right)$ using total communication $\left(\frac{n+s}{R^2}\right) \cdot \text{polylog}(nsT)$, improving on prior bounds (Jia et al., 2025), which only handle $p = 1$. A key technical contribution lies in our algorithmic framework for $\ell_p$ losses, which are significantly more nuanced than $\ell_1$. By embedding $\ell_p$ losses into an $\ell_\infty$ structure using random exponential scalings, we reduce the problem to tracking the largest scaled contribution to the loss across all experts. To control the unbounded variance inherent in exponential random variables, we introduce a geometric mean estimator over independent scalings, yielding an unbiased estimate with provably bounded variance. This estimator is both an algorithmic and a technical novelty, and it may have broader applicability beyond distributed online learning with experts. While the geometric mean is highly effective, alternative variance-reduction techniques could also be explored, representing a promising and fruitful direction for future work. Finally, our results highlight the fundamental tradeoff between communication and regret in distributed online learning with general $\ell_p$ losses. Extending these techniques to other structured loss functions, such as submodular objectives or $\ell_\infty$ (max) losses, presents an exciting avenue for further advancing distributed sequential decision-making.

ACKNOWLEDGMENTS

David P. Woodruff is supported in part Office of Naval Research award number N000142112647, and a Simons Investigator Award. Samson Zhou is supported in part by NSF CCF-2335411. Samson Zhou gratefully acknowledges funding provided by the Oak Ridge Associated Universities (ORAU) Ralph E. Powe Junior Faculty Enhancement Award.

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

# A  RELATED WORKS

In this section, we provide a brief outline of previous literature on the learning with experts problem. In Appendix A.1, we first describe a number of significant contributions to the problem in the central setting, where all $n$ experts and their predictions are available to an algorithm, and there are no restrictions on the memory bounds of the algorithm. Since all information is central, there are also no communication restrictions, unlike the setting we study. We then outline previous works for the learning with experts problem in big data models in Appendix A.2.

## A.1  CLASSIC LEARNING WITH EXPERTS

The problem of online learning with experts has a rich history with extensive connections to reinforcement learning and optimization. Early attempts to understand the problem first assumed the existence of a "perfect expert" – one whose predictions are always correct. A folklore "halving" algorithm achieves at most $\log_2 n$ mistakes given $n$ experts in this setting by retaining the set of experts who have not yet made a mistake and predicts based on a majority vote, eliminating all disagreeing experts for each incorrect prediction.

**Randomized weighted majority.**  However, the assumption of a perfect expert is often unrealistic. The deterministic weighted majority algorithm relaxes this by assigning weights to each expert, predicting based on the weighted majority, and reducing the weights of incorrect experts multiplicatively by $(1 - \varepsilon)$ for fixed any parameter $\varepsilon \in (0, 1)$. Quantitatively, the number of mistakes by the deterministic weighted majority algorithm is at most $2(1 + \varepsilon)m^* + \mathcal{O}\left(\frac{\log n}{\varepsilon}\right)$, where $m^*$ is the number of mistakes made by the best expert (Littlestone & Warmuth, 1994). In fact, a simple example with two experts who always pick different outcomes at each time can force any deterministic algorithm to be incorrect on every time, while each expert is only incorrect on half of the times. Hence, no deterministic algorithm can do better than a multiplicative-factor of two compared to the best expert, i.e., no deterministic algorithm can achieve sublinear regret.

To overcome the worst-case limitations of deterministic algorithms, Littlestone and Warmuth introduced the randomized weighted majority (Littlestone & Warmuth, 1994). Instead of predicting based on a strict weighted majority, randomized weighted majority predicts according to a probability distribution over the experts, where the probability of choosing an expert is proportional to their weight. When an expert makes a mistake, their weight is again decreased multiplicatively by $(1 - \varepsilon)$, so that the algorithm makes a total of $(1 + \varepsilon)m^* + \mathcal{O}\left(\frac{\log n}{\varepsilon}\right)$ mistakes, which translates to $\mathcal{O}\left(\sqrt{T \log n}\right)$ regret for $\varepsilon = \Theta\left(\sqrt{\frac{\log n}{T}}\right)$ and is known to be asymptotically optimal (Cover, 1966).

**Other randomized variants.**  Besides these fundamental approaches, a variety of other algorithms have been developed, each with separate benefits. For example, the well-known Multiplicative Weights Update (MWU) framework generalizes the weight update rule and can accommodate various loss functions (Brown, 1951). Notably, MWU has applications in boosting algorithms like AdaBoost (Freund & Schapire, 1997) and approximately solving zero-sum games (Freund & Schapire, 1999), and it can also efficiently approximate a wide class of linear and semi-definite programs (Clarkson, 1995), leading to fast approximations for various NP-complete problems such as the traveling salesperson problem, scheduling problems, and multi-commodity flow (Plotkin et al., 1995; Garg & Könemann, 2007).

Building upon the "Follow the Leader" principle, which naïvely selects the expert with the lowest cumulative loss so far, Follow the Perturbed Leader (FTPL) introduces an element of randomness by adding perturbations to the cumulative losses of experts before making a selection (Kalai & Vempala, 2005). This randomization proves beneficial in avoiding worst-case scenarios often encountered by deterministic algorithms, leading to asymptotically optimal regret bounds similar to randomized weighted majority and MWU. Moreover, for some structured problems, FTPL is more computationally efficient than MWU. In contrast, Follow the Regularized Leader (FTRL) incorporates a regularization term into the decision process (McMahan, 2011), which penalizes drastic

changes, promoting stability against noise, and offers flexibility to incorporate prior knowledge by choosing the regularization function.

## A.2 LEARNING WITH EXPERTS IN BIG DATA SETTINGS

We remark that all of the algorithms discussed in Appendix A.1 require storing the performance of all experts across all times throughout the duration of the algorithm. However, in many real-world applications, the number of experts $n$ and the time horizon $T$ can be substantial, so that storing and processing all expert performances can be computationally demanding, either requiring significant memory or communication. As a result, a line of recent work has focused on developing online learning with experts algorithms that operate with memory usage that is sublinear in the number of experts $n$.

**Learning with experts on data streams.** Srinivas et al. (2022) showed that any algorithm using $S$ total space must achieve regret $\tilde{\mathcal{O}}\left(\sqrt{\frac{nT}{S}}\right)$, implying that in order to achieve the optimal $\mathcal{O}\left(\sqrt{T \log n}\right)$ regret, the space used must be roughly linear, i.e., $S = \tilde{\mathcal{O}}(n)$. Srinivas et al. (2022) also initially proposed an algorithm for online learning with experts in the random-order model, achieving $\tilde{\mathcal{O}}\left(\frac{nT}{R}\right)$ space for regret $R$. Subsequent work by Peng & Zhang (2023); Peng & Rubinstein (2023) refined these bounds, culminating in Peng & Rubinstein (2023)'s algorithm with $\tilde{\mathcal{O}}\left(\sqrt{nT}\right)$ regret using polylogarithmic space in the arbitrary-order model.

**Multi-armed bandits.** Space complexity has also been studied for multi-armed bandits, a related reinforcement learning problem with arms having fixed reward distributions, but only feedback for a single arm is given on each time. Liau et al. (2018) showed that achieving near-optimal regret requires only constant space. However, this approach is not immediately applicable to the our setting, due to the adversarial nature of expert predictions, unlike the static rewards in bandits, and because expert algorithms observe all "arms" each day. Thus, expert and bandit algorithms are not directly comparable. Assadi & Wang (2020) showed that $\Omega(k)$ space is necessary to identify the top $k$ arms in a streaming bandit model. These results indicate that the experts problem is inherently harder than bandits, as low-regret bandit solutions can have constant space complexity.

**Learning in streams.** There is also significant research on the tradeoffs between space and sample complexity for statistical learning and estimation in the i.i.d. streaming model, including studies on matrix row inference (Raz, 2017; Garg et al., 2018; 2019) and parity learning (Kol et al., 2017; Raz, 2019). Another line of active work studies specific streaming learning problems such as correlation finding (Dagan & Shamir, 2018), collision probability estimation, graph connectivity, and rank estimation (Crouch et al., 2016). However, our work on the experts problem does not assume any data distribution and focuses on prediction rather than inference. Thus, these works differs significantly from our goal of proving general space complexity bounds for expert algorithms while maintaining competitive regret against the best expert.

# B MISSING PROOFS

## B.1 MISSING PROOFS FROM SECTION 2

**Lemma 2.3.** *[Max stability of exponential random variables] Let* $X_i = \frac{f_i}{e_i^{1/p}}$ *for* $i \in [n]$, *where* $e_1, \ldots, e_n$ *are independent exponential random variables with rate* $1$. *Then* $\max_{i \in [n]} \frac{f_i}{e_i^{1/p}}$ *is distributed as* $\frac{\|f\|_p}{e^{1/p}}$, *where* $e$ *is an exponential random variable with rate* $1$.

*Proof.* Let $X = \max_{i \in [n]} \frac{f_i}{e_i^{1/p}}$. Then for any $t > 0$, we have

$$\mathbf{Pr}\left[X \le t\right] = \mathbf{Pr}\left[X^p \le t^p\right]$$

$$= \prod_{i=1}^{n} \mathbf{Pr}\left[\frac{f_i^p}{e_i} \le t^p\right]$$

$$= \prod_{i=1}^{n} \exp\left(-t^p \cdot f_i^p\right)$$

$$= \exp\left(-t^p \cdot \|f\|_p^p\right).$$

By comparison, if $Y = \frac{\|f\|_p}{e^{1/p}}$, where $e$ is an exponential random variable with rate 1, then

$$\mathbf{Pr}\left[Y \le t\right] = \mathbf{Pr}\left[Y^p \le t^p\right] = \exp\left(-t^p \cdot \|f\|_p^p\right).$$

Hence, $\max_{i \in [n]} \frac{f_i}{e_i^{1/p}}$ follows the same distribution as $\frac{\|f\|_p}{e^{1/p}}$. $\qquad\square$

## B.2 MISSING PROOFS FROM SECTION 3

**Lemma 3.2.** *[Expectation and variance of geometric mean estimator] Let $p > 0$ be fixed and $R \ge 3p$ be an integer. Let $e_1, \ldots, e_B$ be independent exponential random variables with rate 1. Let $Z_b = \frac{1}{e_b^{1/p}}$ for all $b \in [B]$, and let $Z = \left(\prod_{b \in [B]} Z_b\right)^{1/B}$ be their geometric mean. Then there exists a universal constant $C_{3.2} \in (0, 2^B]$ such that $\mathbb{E}\left[Z\right] = C_{3.2}$ and $\mathbb{E}\left[Z^2\right] \le 3^B$.*

*Proof.* The probability density function for an exponential random variable with rate 1 is $f(x) = e^{-x}$. Thus by Fact 3.1, the probability density function for each term $Z_b$ is $px^{-p-1} \cdot e^{-x^{-p}}$ and similarly, the probability density function for each term $Z_b^{1/B}$ in the geometric mean is $p(x) = Bpx^{-Bp-1} \cdot e^{-x^{-Bp}}$. Therefore, we have $\mathbf{Pr}\left[Z_b^{1/B} > t\right] \le \frac{1}{t^{Bp}}$, so that for $B = \left\lceil \frac{3}{p} \right\rceil$.

$$\mathbb{E}\left[Z_b^{1/B}\right] = \int_0^{\infty} \mathbf{Pr}\left[Z_b^{1/B} > t\right] dt$$

$$= \int_0^1 \mathbf{Pr}\left[Z_b^{1/B} > t\right] dt + \int_1^{\infty} \mathbf{Pr}\left[Z_b^{1/B} > t\right] dt$$

$$\le 1 + \int_1^{\infty} \frac{1}{t^{Bp}} dt$$

$$\le 1 + \int_1^{\infty} \frac{1}{t^3} dt \le 2.$$

Since $\mathbb{E}\left[Z\right] = \prod_{b=1}^{B} \mathbb{E}\left[Z_b^{1/B}\right]$, then we have $\mathbb{E}\left[z\right] \le 2^B$. Moreover, the probability density function for each term $Z_b^{1/B}$ in the geometric mean satisfies $p(x) = Bpx^{-Bp-1} \cdot e^{-x^{-Bp}} > Bp \cdot e^{-2^{Bp}}$ for all $x \in \left[\frac{1}{2}, 1\right]$.

$$\mathbb{E}\left[Z_b^{1/B}\right] = \int_0^{\infty} p(t) \cdot t \, dt$$

$$> \int_{1/2}^1 p(t) \cdot t \, dt$$

$$\ge \frac{1}{2} \cdot \min_{t \in \left[\frac{1}{2}, 1\right]} p(t) \cdot t$$

$$\ge \frac{1}{2} \cdot \min_{t \in \left[\frac{1}{2}, 1\right]} Bp \cdot e^{-2^{Bp}} \cdot \frac{1}{2} = \Omega(1),$$

so that $\mathbb{E}\left[Z_b^{1/B}\right] = \Omega(1)$ is bounded away from 0. Thus, $\mathbb{E}\left[Z\right]$ equals some constant $C_{3.2} \in (0, 2^B]$.

By similar reasoning, we have

$$\mathbb{E}[Z_b^{2/R}] = \int_0^{\infty} \mathbf{Pr}\left[Z_b^{2/R} > t\right] dt$$

$$\le \int_0^1 \mathbf{Pr}\left[Z_b^{2/R} > t\right] dt + \int_1^{\infty} \mathbf{Pr}\left[Z_b^{2/R} > t\right] dt$$

$$\leq 1 + \int_1^\infty \frac{1}{t^{Bp/2}}\, dt$$

$$\leq 1 + \int_1^\infty \frac{1}{t^{3/2}}\, dt \leq 3.$$

Hence, we have $\mathbb{E}\left[Z^2\right] = \prod_{b=1}^B \mathbb{E}\left[Z_b^{2/R}\right]$, so that $\mathbb{E}\left[Z^2\right] \in [0, 3^B]$. $\qquad\square$

**Lemma 3.3.** *Suppose $\mathbb{E}\left[\widehat{s_i(t)}\right] = L_i(t)$ and this random variable has second moment at most $\rho$ for all $i \in [n]$ and $t \in [T]$. Suppose the multiplicative weights update (MWU) algorithm is executed with loss sequences $\widehat{s_i(t)}$ instead of $L_i(t)$ for all $i \in [n]$ and $t \in [T]$. Then the sequence of resulting choices achieves expected regret $\mathcal{O}\left(\sqrt{\frac{\rho \log n}{T}}\right)$ on the sequence of losses $\{L_i(t)\}_{i \in [n], t \in [T]}$.*

*Proof.* For each $t \in [T]$, let $w_t = (w_{t,1}, \ldots, w_{t,n})$ be the vector of expert weights maintained by MWU. Then the probability distribution $p_t$ satisfies $p_t = \frac{w_t}{\sum_{j=1}^n w_{t,j}}$. After observing the loss estimates $\widehat{s_i(t)} \geq 0$, the weights are updated as $w_{t+1,i} = w_{t,i} \exp(-\eta \widehat{s_i(t)})$. By assumption, we have that $\widehat{s_i(t)}$ is an unbiased estimate of $L_i(t)$ independent of MWU, and thus for every realization of $p_t$ and each $i, t$, we have

$$\mathbb{E}\left[\widehat{s_i(t)} \mid p_t\right] = L_i(t), \qquad \mathbb{E}\left[\widehat{s_i(t)}^2 \mid p_t\right] \leq \rho.$$

For $y \geq 0$, $e^{-y} \leq 1 - y + y^2$, so

$$\mathbb{E}\left[e^{-\eta \widehat{s_i(t)}} \mid p_t\right] \leq 1 - \eta L_i(t) + \eta^2 \rho.$$

For $W_t = \sum_i w_{t,i}$ and $L_t = (L_1(t), \ldots, L_n(t))$ and summing over $i$ with weights $w_{t,i}$,

$$\mathbb{E}\left[W_{t+1} \mid \mathcal{F}_t\right] = \sum_i w_{t,i} \mathbb{E}\left[e^{-\eta \widehat{s_i(t)}} \mid p_t\right] \leq W_t(1 - \eta\, p_t \cdot L_t + \eta^2 \rho).$$

Iterating and taking expectations,

$$\mathbb{E}\left[W_{T+1}\right] \leq n \prod_{t=1}^T (1 - \eta\, \mathbb{E}\left[p_t \cdot L_t\right] + \eta^2 \rho).$$

For any expert $i^*$, $w_{T+1,i^*} = \exp(-\eta \sum_t \widehat{s_{i^*}(t)}) \leq W_{T+1}$, hence

$$\mathbb{E}\left[e^{-\eta \sum_t \widehat{s_{i^*}(t)}}\right] \leq n \prod_{t=1}^T (1 - \eta\, \mathbb{E}\left[p_t \cdot L_t\right] + \eta^2 \rho).$$

Taking logarithms and using $\ln(1 + u) \leq u$,

$$\ln \mathbb{E}\left[e^{-\eta \sum_t \widehat{s_{i^*}(t)}}\right] \leq \ln n - \eta \sum_t \mathbb{E}\left[p_t \cdot L_t\right] + \eta^2 \rho T.$$

By Jensen's inequality,

$$-\eta \sum_t L_{i^*}(t) = \ln e^{-\eta \sum_t L_{i^*}(t)} \leq \ln \mathbb{E}\left[e^{-\eta \sum_t \widehat{s_{i^*}(t)}}\right].$$

Combining the two inequalities gives

$$-\eta \sum_t L_{i^*}(t) \leq \ln n - \eta \sum_t \mathbb{E}\left[p_t \cdot L_t\right] + \eta^2 \rho T,$$

which rearranges to

$$\sum_t \mathbb{E}\left[p_t \cdot L_t\right] - \sum_t L_{i^*}(t) \leq \frac{\ln n}{\eta} + \eta \rho T.$$

Thus, the loss of MWU is at most $\frac{\ln n}{\eta} + \eta \rho T$. Choosing $\eta = \sqrt{\frac{\ln n}{\rho T}}$ and normalizing by time horizon $T$ gives a regret bound of at most $\mathcal{O}\left(\sqrt{\frac{\rho \log n}{T}}\right)$. $\qquad\square$

**Lemma B.1.** *Let $q_i^{(b)}(j,t)$ for $j \in [s]$ be as defined in Lemma 2.3, and suppose $L_i(t) \geq s^{1/p}$. Then*

$$\mathbf{Pr}\left[\max_{j \in [s]} q_i^{(b)}(j,t) < \frac{100 \log(nsT)}{s^{1/p}}\right] \leq \frac{1}{\text{poly}(nST)}.$$

*Proof.* By Lemma 2.3 and Lemma 3.2, $\max_{j \in [s]} q_i^{(b)}(j,t)$ is distributed as $C_{3.2} \, e^{1/p} L_i(t)$, where $e$ is an exponential random variable with rate 1. Thus

$$\mathbf{Pr}\left[\max_{j \in [s]} q_i^{(b)}(j,t) < \frac{100 \log(nsT)}{s^{1/p}}\right] = \mathbf{Pr}\left[C_{3.2} \, e^{1/p} L_i(t) < \frac{100 \log(nsT)}{s^{1/p}}\right]$$

$$= \mathbf{Pr}\left[e > \left(\frac{C_{3.2} \, s^{1/p}}{100 \, L_i(t) \, \log(nsT)}\right)^p\right].$$

Since $L_i(t) \geq s^{1/p}$, we have

$$\mathbf{Pr}\left[e > \left(\frac{C_{3.2} \, s^{1/p}}{100 \, L_i(t) \, \log(nsT)}\right)^p\right] \leq \exp\left(-\frac{(100 \log(nsT))^p}{C_{3.2}^p}\right) \leq \frac{1}{\text{poly}(nsT)}.$$

$\square$

**Lemma 3.4.** *Suppose $\ell_i(j,t) \in [1,5]$ for all $t \in [T]$, $i \in [n]$ and $j \in [s]$. Then the expected regret of Algorithm 2 is at most $\mathcal{O}\left(s^{1/p}\sqrt{\frac{\log n}{T}}\right)$.*

*Proof.* Since the loss on each server is in the range $[1,5]$, then the loss $L_i(t)$ on each expert on each day is at most $\mathcal{O}\left(s^{1/p}\right)$. By Lemma 2.3, we have that $\widehat{s_i(t)}$ has the same distribution as $\frac{L_i(t)}{e^{1/p}}$ for an exponential random variable $e$. By Lemma 3.2, $\widehat{s_i(t)}$ is an unbiased estimate of $L_i(t) + \frac{1}{\text{poly}(nT)}$ with second moment $\mathcal{O}\left(s^{1/p}\right)$. Thus by Lemma 3.3, the expected regret of Algorithm 2 is at most

$$\mathcal{O}\left(s^{1/p}\sqrt{\frac{\log n}{T}}\right) + \frac{1}{\text{poly}(nT)} \cdot T = \mathcal{O}\left(s^{1/p}\sqrt{\frac{\log n}{T}}\right). \qquad \square$$

**Lemma 3.5.** *Suppose $\ell_i(j,t) \in [1,5]$ for all $t \in [T]$, $i \in [n]$ and $j \in [s]$. Then with high probability, the total communication of Algorithm 2 is at most $\mathcal{O}\left(sT\right) + nT \cdot \text{polylog}(nsT)$ bits.*

*Proof.* Consider Algorithm 2. Observe that at time $t \in [T]$, server $j \in [s]$ will communicate an expert $i \in [n]$ only if $q_i^{(b)}(j,t) \geq \frac{s^{1/p}}{100 \log(nsT)}$. By assumption, we have $\ell_i(j,t) \leq 5$ for all $t \in [T]$, $i \in [n]$ and $j \in [s]$. Since $q_i^{(b)}(j,t) = \frac{\ell_i(j,t)}{(e_i^{(b)}(j,t))^{1/p}}$ for $b \in [B]$, then for a fixed $b \in [B]$, server $j \in [s]$ will use communication only if $(e_i^{(b)}(j,t))^{1/p} \leq \frac{500 \log(nsT)}{s^{1/p}}$ or equivalently, if $e_i^{(b)}(j,t) \leq \frac{500^p \log^p(nT)}{s}$ Since $(e_i^{(b)}(j,t))$ is an independent exponential random variable with rate 1, then we have

$$\mathbf{Pr}\left[(e_i^{(b)}(j,t))^{1/p} \leq \frac{500 \log(nsT)}{s^{1/p}}\right] \lesssim \frac{\log^p(nT)}{s}.$$

Now for a fixed $i \in [n]$, let $Y_1, \ldots, Y_S$ denote indicator random variables for whether $q_i^{(b)}(j,t)$ triggers a message from the server to the coordinator. In other words, $Y_j = 1$ if server $j$ sends a message due to $i$ and $Y_j = 0$ otherwise. Then we have

$$\mathbb{E}\left[Y_1 + \ldots + Y_S\right] \lesssim \log^p(nsT).$$

By standard Chernoff bounds, c.f., Theorem 2.1, we have that $\mathbf{Pr}\left[Y_1 + \ldots + Y_s \gtrsim \log^{p+1}(nsT)\right] \leq \frac{1}{(nsT)^{10}}$. By a union bound over all $b \in [B]$, $i \in [n]$, $t \in [T]$, it follows that the total number of samples sent to the coordinator is at most $nT \cdot \text{polylog}(nsT)$. On the other hand, the coordinator needs to sync with every server for all $t \in [T]$, inducing $\mathcal{O}\left(sT\right)$ communication. Hence, with high probability, the overall communication is at most $\mathcal{O}\left(sT\right) + nT \cdot \text{polylog}(nsT)$. $\square$

## B.3 MISSING PROOFS FROM SECTION 4

**Fact 4.1.** *Let $e$ be an exponential random variable with rate $1$. Then for all $x > 2$, we have $\mathbf{Pr}\left[\frac{1}{e} \in (x, 2x]\right] \in \left[\frac{1}{4x}, \frac{1}{2x}\right]$.*

*Proof.* By Fact 3.1, the probability density function for $\frac{1}{e}$ is $p(x) = \frac{1}{x^2} \cdot e^{-1/x}$. We have

$$\mathbf{Pr}\left[\frac{1}{e} \in (x, 2x]\right] = \int_x^{2x} p(x)\, dx = \int_x^{2x} \frac{1}{x^2} \cdot e^{-1/x}\, dx.$$

Note that for $x > 2$, we have $e^{1/x} \in \left(\frac{1}{2}, 1\right]$. Hence,

$$\mathbf{Pr}\left[\frac{1}{e} \in (x, 2x]\right] \leq \int_x^{2x} \frac{1}{x^2}\, dx = \frac{1}{x} - \frac{1}{2x} = \frac{1}{2x},$$

and

$$\mathbf{Pr}\left[\frac{1}{e} \in (x, 2x]\right] \geq \int_x^{2x} \frac{1}{2} \frac{1}{x^2}\, dx = \frac{1}{2x} - \frac{1}{4x} = \frac{1}{4x}.$$

$\square$

**Lemma 4.2.** *We have $\mathbb{E}\left[\widehat{s_i(t)}\right] = L_i(t) + \frac{1}{\mathrm{poly}(nT)}$ and $\mathbb{E}\left[(\widehat{s_i(t)})^2\right] \leq \mathcal{O}\left(TR^2\right) \cdot 3^B \cdot (L_i(t))^2$.*

*Proof.* Let $\varrho$ be the probability of sampling each time $t \in [T]$. Observe that with probability $1 - \varrho$, we have $\widehat{s_i(t)} = 0$. Otherwise, we continue as before.

Let $e, e_1, \ldots, e_B$ be independent random variables. Then by Lemma 2.3, we have

$$
\begin{aligned}
\mathbb{E}\left[\widehat{s_i(t)}\right] &= \varrho \cdot \frac{1}{\varrho} \cdot \mathbb{E}\left[\prod_{b=1}^B \left(\max_{j \in [s]} q_i^{(b)}(j, t)\right)^{1/B}\right] \\
&= \varrho \cdot \frac{1}{\varrho} \cdot \mathbb{E}\left[\prod_{b=1}^B \left(\frac{L_i(t)}{C_{3.2} e_b^{1/p}}\right)^{1/B}\right] \\
&= \varrho \cdot \frac{1}{\varrho} \cdot \prod_{b=1}^B \mathbb{E}\left[\left(\frac{L_i(t)}{C_{3.2} e^{1/p}}\right)^{1/B}\right] \\
&= \frac{L_i(t)}{C_{3.2}} \cdot \left(\mathbb{E}\left[\frac{1}{e^{1/Bp}}\right]\right)^B \\
&= L_i(t),
\end{aligned}
$$

where the last equality follows by the definition of $C_{3.2} = \left(\mathbb{E}\left[\frac{1}{e^{1/Bp}}\right]\right)^B$ in Lemma 3.2.

Similarly, we have

$$
\begin{aligned}
\mathbb{E}\left[(\widehat{s_i(t)})^2\right] &= p \cdot \frac{1}{\varrho^2} \cdot \mathbb{E}\left[\prod_{b=1}^B \left(\frac{(L_i(t))^2}{(C_{3.2})^2 e_b^{2/p}}\right)^{1/B}\right] \\
&= \frac{1}{\varrho} \cdot \prod_{b=1}^B \mathbb{E}\left[\left(\frac{(L_i(t))^2}{(C_{3.2})^2 e_b^{2/p}}\right)^{1/B}\right] \\
&= \frac{1}{\varrho} \cdot \prod_{b=1}^B \frac{(L_i(t))^2}{(C_{3.2})^2} \cdot \mathbb{E}\left[\frac{1}{e_b^{2/Bp}}\right]^B.
\end{aligned}
$$

By Lemma 3.2, we have $C_{3.2} = \Theta(1)$ and $\mathbb{E}\left[\frac{1}{e_b^{2/Bp}}\right]^B \leq 3^B$. Moreover, we set $\varrho = \frac{1}{TR^2}$, so that $\frac{1}{\varrho} = TR^2$. Therefore,

$$\mathbb{E}\left[(\widehat{s_i(t)})^2\right] \leq TR^2 \cdot 3^B \cdot \mathcal{O}(1) \cdot (L_i(t))^2,$$

as desired. □

**Lemma 4.3.** *Suppose $\ell_i(j,t) \in [1,5]$ for all $t \in [T]$, $i \in [n]$ and $j \in [s]$. The expected regret of Algorithm 3 is at most $\mathcal{O}\left(Rs^{1/p}\sqrt{\log n}\right)$.*

*Proof.* Since the loss on each server is in the range $[1,5]$, then the loss $L_i(t)$ on each expert on each day is at most $\mathcal{O}\left(s^{1/p}\right)$. By Lemma 5.1, we have that $\mathbb{E}\left[\widehat{s_i(t)}\right] = L_i(t) + \frac{1}{\text{poly}(nT)}$ and $\mathbb{E}\left[(\widehat{s_i(t)})^2\right] = \mathcal{O}\left(TR^2\right) \cdot 3^B \cdot (L_i(t))^2$. Thus by Lemma 3.3, the expected regret of Algorithm 3 is at most $\mathcal{O}\left(Rs^{1/p}\sqrt{3^B \log n}\right) + \frac{1}{\text{poly}(nT)} \cdot T = \mathcal{O}\left(Rs^{1/p}\sqrt{3^B \log n}\right)$. □

**Lemma 4.4.** *Suppose $\ell_i(j,t) \in [1,5]$ for all $t \in [T]$, $i \in [n]$ and $j \in [s]$. Then with high probability, the total communication is at most $\left(\frac{n+s}{R^2}\right) \cdot \text{polylog}(nsT)$ bits.*

*Proof.* Consider Algorithm 3. Observe that at time $t \in [T]$, server $j \in [s]$ will communicate an expert $i \in [n]$ only if first it is sampled with probability $\varrho$ and then $q_i^{(b)}(j,t) \geq \frac{s^{1/p}}{100 \log(nsT)}$. By assumption, we have $\ell_i(j,t) \leq 5$ for all $t \in [T]$, $i \in [n]$ and $j \in [s]$. Since $q_i^{(b)}(j,t) = \frac{\ell_i(j,t)}{(e_i^{(b)}(j,t))^{1/p}}$ for $b \in [B]$, then for a fixed $b \in [B]$, server $j \in [s]$ will use communication only if $(e_i^{(b)}(j,t))^{1/p} \leq \frac{500 \log(nsT)}{s^{1/p}}$ or equivalently, if $e_i^{(b)}(j,t) \leq \frac{500^p \log^p(nT)}{s}$ Since $(e_i^{(b)}(j,t))$ is an independent exponential random variable with rate 1, then we have

$$\mathbf{Pr}\left[(e_i^{(b)}(j,t))^{1/p} \leq \frac{500 \log(nsT)}{s^{1/p}}\right] \lesssim \frac{\log^p(nT)}{s}.$$

Now for a fixed $i \in [n]$, let $Y_1, \ldots, Y_S$ denote indicator random variables for whether $q_i^{(b)}(j,t)$ triggers a message from the server to the coordinator. In other words, $Y_j = 1$ if server $j$ sends a message due to $i$ and $Y_j = 0$ otherwise. Then for each $j \in [s]$, we have

$$\mathbb{E}[Y_j] \lesssim \varrho \cdot \frac{\log^p(nT)}{s},$$

since the event of probability $\varrho$ must occur first. Then by linearity of expectation, we have

$$\mathbb{E}[Y_1 + \ldots + Y_s] \lesssim \varrho \cdot \log^p(nsT).$$

Similarly, we have that over all times and over all experts, the total expected amount of communication due to these events is at most $\mathcal{O}\left(\varrho \cdot nT \cdot \log^p(nsT)\right)$. Recall that Algorithm 3 sets $\varrho = \frac{1}{R^2 T}$. By standard Chernoff bounds, c.f., Theorem 2.1, it follows that the total number of samples sent to the coordinator is at most $\frac{n}{R^2} \cdot \text{polylog}(nsT)$, with high probability. On the other hand, the coordinator needs to sync with every server for all selected days, in case the server does not communicate anything. This process induces $\frac{s}{R^2} \cdot \text{polylog}(nsT)$ communication with high probability, by a similar Chernoff bound argument. Therefore, with high probability, the total communication is at most $\left(\frac{n+s}{R^2}\right) \cdot \text{polylog}(nsT)$, as claimed. □

### B.4 Missing Proofs from Section 5

**Lemma 5.1.** *Suppose $q_i^{(b)}(j,t) \geq \frac{1}{100\sqrt{T}}$ for all $b \in [B]$. Then we have $\mathbb{E}\left[\widehat{s_i(t)}\right] = L_i(t) + \frac{1}{\text{poly}(nT)}$ and $\mathbb{E}\left[(\widehat{s_i(t)})^2\right] \leq s^{2/p} \cdot R^2 T$.*

*Proof.* Let $a^* \geq 0$ be the smallest integer such that $\widehat{s_i(t)} \geq \left(\frac{s}{2^{a^*}}\right)^{1/p}$. Observe that we have $a^* \leq a$ with probability $\varrho_{a^*} := \frac{\varrho}{2^{a^*}}$. Moreover, we require that $a^* \leq a$ to set $w_i(t) = w_i(t-1) + 2^{a^*} \cdot \widehat{s_i(t)}$. Consider a hypothetical process $\mathcal{P}$ where $q_i^{(b)}(j,t)$ is reported by all servers if $a^* \leq a$. We shall ultimately show that $\mathcal{P}$ holds with high probability so that the expected regret is also slightly affected by the actual process.

Now, in this process, $\widehat{s_i(t)}$ is nonzero with probability $\varrho_{a*}$. Recall that $B = \left\lceil \frac{3}{p} \right\rceil$. Then by Lemma 2.3,

$$
\begin{aligned}
\mathbb{E}\left[\widehat{s_i(t)}\right] &= \mathbb{E}\left[\varrho_{a*} \cdot \frac{1}{\varrho_{a^*}} \cdot \prod_{b=1}^{B}\left(\max_{j\in[s]} q_i^{(b)}(j,t)\right)^{1/B}\right] \\
&= \prod_{b=1}^{B}\mathbb{E}\left[\left(\max_{j\in[s]} q_i^{(b)}(j,t)\right)^{1/B}\right] \\
&= \left(\mathbb{E}\left[\left(\max_{j\in[s]} q_i^{(b)}(j,t)\right)^{1/B}\right]\right)^{B} \\
&= \left(\mathbb{E}\left[\left(\frac{L_i(t)}{C_{3.2}e^{1/p}}\right)^{1/B}\right]\right)^{B} \\
&= \frac{L_i(t)}{C_{3.2}} \cdot \left(\mathbb{E}\left[\frac{1}{e^{1/Bp}}\right]\right)^{B} \\
&= L_i(t),
\end{aligned}
$$

where the last equality follows by the definition of $C_{3.2} = \left(\mathbb{E}\left[\frac{1}{e^{1/Bp}}\right]\right)^{B}$ in Lemma 3.2.

Similarly,

$$
\begin{aligned}
\mathbb{E}\left[(\widehat{s_i(t)})^2\right] &= \mathbb{E}\left[\varrho_{a^*} \cdot \frac{1}{\varrho_{a^*}^2} \cdot \prod_{b=1}^{B}\left(\max_{j\in[s]} q_i^{(b)}(j,t)\right)^{2/B}\right] \\
&= \mathbb{E}\left[\frac{1}{\varrho_{a^*}}\prod_{b=1}^{B}\left(\max_{j\in[s]} q_i^{(b)}(j,t)\right)^{2/B}\right].
\end{aligned}
$$

We have

$$
\prod_{b\in[B]}\left(\max_{j\in[s]}\widehat{q_i^{(b)}(j,t)}\right)^{1/B} \in \left[\left(\frac{s}{2^{a^*}}\right)^{1/p}, \left(\frac{s}{2^{a^*-1}}\right)^{1/p}\right].
$$

We also have $\varrho_{a^*} = \frac{\varrho}{2^{a^*}}$. Therefore,

$$
\begin{aligned}
\mathbb{E}\left[(\widehat{s_i(t)})^2\right] &= \mathbb{E}\left[\frac{1}{p_{a^*}} \cdot \prod_{b=1}^{B}\left(\max_{j\in[s]} q_i^{(b)}(j,t)\right)^{2/B}\right] \\
&\leq \mathbb{E}\left[\frac{2^{a^*}}{\varrho} \cdot \left(\frac{s}{2^{a^*}}\right)^{2/p}\right] \\
&\leq \frac{(2^{a^*})^{1-2/p} \cdot s^{2/p}}{\varrho}.
\end{aligned}
$$

Now, for $p \leq 2$, we have $(2^{a^*})^{1-2/p} \leq 1$ and $\varrho = \frac{1}{R^2T} \cdot \max(s^{1-2/p}, 1) = \frac{1}{R^2T}$, so that $\mathbb{E}\left[(\widehat{s_i(t)})^2\right] \leq R^2T$.

Similarly, for $p > 2$, we have $\varrho = \frac{1}{R^2T} \cdot \max(s^{1-2/p}, 1) = \frac{s^{1-2/p}}{R^2T}$, so that

$$
\mathbb{E}\left[(\widehat{s_i(t)})^2\right] \leq \frac{(2^{a^*})^{1-2/p} \cdot s^{2/p} \cdot R^2T}{s^{1-2/p}}.
$$

Since $2^{a^*} \leq s$, then it follows that $\mathbb{E}\left[(\widehat{s_i(t)})^2\right] \leq s^{2/p} \cdot R^2T$, as desired. $\qquad\square$

We upper bound the regret of Algorithm 4.

**Lemma 5.2.** *Suppose we have $\ell_i(j,t) \leq 1$ for all $t \in [T]$. The expected regret of Algorithm 4 is at most $\mathcal{O}\left(Rs^{1/p}\sqrt{\log n}\right)$.*

*Proof.* Since the loss on each server is at most 1, then the loss $L_i(t)$ on each expert on each day is at most $\mathcal{O}\left(s^{1/p}\right)$. By Lemma 5.1, we have that $\mathbb{E}\left[\widehat{s_i(t)}\right] = L_i(t) + \frac{1}{\text{poly}(nT)}$ and $\mathbb{E}\left[(\widehat{s_i(t)})^2\right] \leq s^{2/p} \cdot R^2 T$. Thus by Lemma 3.3, the expected regret of Algorithm 4 is at most $\mathcal{O}\left(Rs^{1/p}\sqrt{\log n}\right) + \frac{1}{\text{poly}(nT)} \cdot T = \mathcal{O}\left(Rs^{1/p}\sqrt{\log n}\right)$. $\qquad\square$

It remains to upper bound the total communication of Algorithm 4.

**Lemma 5.3.** *Suppose we have $\ell_i(j, t) \leq 1$ for all $t \in [T]$. Then with high probability, the total communication for Algorithm 4 is at most $\left(\frac{n+s}{R^2}\right) \cdot \max(s^{1-2/p}, 1) \cdot \text{polylog}(nsT)$ bits.*

*Proof.* Consider Algorithm 4 and a fixed $a \in [A]$. At time $t \in [T]$, server $j \in [s]$ will communicate with expert $i \in [n]$ only if it is first selected with probability $\frac{\varrho}{2^a}$, and then $q_i^{(b)}(j, t) \geq \frac{s^{1/p}}{100 \cdot (2^a)^{1/p} \log(nsT)}$. By assumption, we have $\ell_i(j, t) \leq 1$ for all $t \in [T]$, $i \in [n]$, and $j \in [s]$. Since $q_i^{(b)}(j, t) = \frac{\ell_i(j,t)}{(e_i^{(b)}(j,t))^{1/p}}$ for $b \in [B]$, for a fixed $b \in [B]$, server $j \in [s]$ will communicate only if $(e_i^{(b)}(j, t))^{1/p} \leq \frac{100 \cdot (2^a)^{1/p} \log(nsT)}{s^{1/p}}$, or equivalently, if $e_i^{(b)}(j, t) \leq \frac{100^p \cdot 2^a \cdot \log^p(nT)}{s}$. Since $(e_i^{(b)}(j, t))$ is an independent exponential random variable with rate 1, we have

$$\mathbf{Pr}\left[(e_i^{(b)}(j, t))^{1/p} \leq \frac{100 \cdot (2^a)^{1/p} \log(nsT)}{s^{1/p}}\right]$$
$$\lesssim \frac{2^a \cdot \log^p(nT)}{s}.$$

Now, for a fixed $i \in [n]$, let $Y_1, \ldots, Y_S$ denote indicator random variables indicating whether $q_i^{(b)}(j, t)$ triggers a message from the server to the coordinator. In other words, $Y_j = 1$ if server $j$ sends a message due to $i$, and $Y_j = 0$ otherwise. Then, for each $j \in [s]$, we have

$$\mathbb{E}[Y_j] \lesssim \varrho \cdot \frac{\log^p(nT)}{s},$$

since the event with probability $\frac{\varrho}{2^a}$ must occur first. By linearity of expectation, we have

$$\mathbb{E}[Y_1 + \ldots + Y_s] \lesssim \varrho \cdot \log^p(nsT).$$

Similarly, over all times and experts, the total expected communication due to these events is at most $\mathcal{O}\left(\varrho \cdot nT \cdot \log^p(nsT)\right)$. Recall that Algorithm 4 sets $\varrho = \frac{1}{R^2 T} \cdot \max(s^{1-2/p}, 1)$. By standard Chernoff bounds, as in Theorem 2.1, the total number of samples sent to the coordinator is at most $\frac{n}{R^2} \cdot \text{polylog}(nsT) \cdot \max(s^{1-2/p}, 1)$, with high probability. On the other hand, the coordinator must sync with every server for all selected days in case the server does not communicate anything. This process induces $\frac{s}{R^2} \cdot \text{polylog}(nsT) \cdot \max(s^{1-2/p}, 1)$ communication with high probability, by a similar Chernoff bound argument. Thus, with high probability, the total communication is at most $\left(\frac{n+s}{R^2}\right) \cdot \max(s^{1-2/p}, 1) \cdot \text{polylog}(nsT)$, as desired. $\qquad\square$

