# OpenReview forum: "Better Bounds for the Distributed Experts Problem"
_ICLR.cc/2026/Conference — ICLR 2026 Poster_

### Official Review · Reviewer_g7FC · 2025-10-20

**Soundness:** 4
**Presentation:** 3
**Contribution:** 2
**Rating:** 2
**Confidence:** 3

**Summary:**

The paper presents a regret bound for a distributed online convex optimisation task.   The setup is that each expert sees a loss vector of length s, where s is the number of servers, and performance is evaluated by the Lp norm of this vector and so requires communication amongst the servers.  Regret is wrt the best expert.  The motivation primarily comes from hyperparameter optimisation, where each expert corresponds to a choice of hyperparameters and each server to a different dataset on which an ML model is evaluated.  The novelty lies mainly in the extension to Lp norms with p>1.

**Strengths:**

The paper is well written, and the analysis appears to be sound (I did not carefully check all of the proofs).

**Weaknesses:**

I found the HPO motivation given for this work unconvincing.  Communication between servers is not really the main bottleneck - sending a few values takes little bandwidth and can be carried out quickly.  Synchronisation seems more of an issue.   It would have been much better to have presented a proper comparison against state of the art hyperparameter optimisation methods in the empirical evaluation section.  That said, the main contribution is probably the theoretical analysis.  I don't see much interesting new maths techniques being introduced, so the contribution is mainly covering Lp loss with p>1.   But online convex optimisation is a mature area these days, with a large literature, and the present work seems like a small extension of previous work and a relatively minor contribution.

**Questions:**

See comments re weaknesses above.

---

> ### Author Response · Authors · 2025-11-21
>
> > I found the HPO motivation given for this work unconvincing. Communication between servers is not really the main bottleneck - sending a few values takes little bandwidth and can be carried out quickly. Synchronisation seems more of an issue. It would have been much better to have presented a proper comparison against state of the art hyperparameter optimisation methods in the empirical evaluation section.
>
> We agree that synchronization can also be a bottleneck in distributed systems. However, communication is also a significant bottleneck, especially when the number of experts and servers is large. Our work focuses on the tradeoff between communication and regret, which is a fundamental question in distributed online learning. We believe that our theoretical results provide insights into this tradeoff, even if communication is not the only bottleneck in practice. Regarding the empirical evaluation, we agree that a comparison against state-of-the-art HPO methods would be interesting. However, our focus is on the theoretical aspects of the distributed experts problem, and our empirical evaluation is intended as a proof of concept.
>
> > That said, the main contribution is probably the theoretical analysis. I don't see much interesting new maths techniques being introduced, so the contribution is mainly covering Lp loss with p>1. But online convex optimisation is a mature area these days, with a large literature, and the present work seems like a small extension of previous work and a relatively minor contribution.
>
> We respectfully disagree that our contribution is minor. While online convex optimization is a mature area, the distributed setting introduces new challenges, especially for general $\ell_p$ losses. Our work is the first to address this problem in the coordinator model, and we introduce novel techniques to handle the non-additivity of $\ell_p$ losses. In particular, the use of exponential random variables and geometric mean estimators to embed $\ell_p$ norms into $\ell_\infty$ norms is a novel contribution in the context of distributed online learning. We believe that these techniques may be of independent interest to other applications.

---

### Official Review · Reviewer_zz4T · 2025-10-30

**Soundness:** 2
**Presentation:** 2
**Contribution:** 2
**Rating:** 2
**Confidence:** 4

**Summary:**

The paper considers learning with experts but in a distributed setting where there are $s$ servers and each of $n$ experts has a 'partial' loss at each server. The total loss of an expert is the $\ell_p$ norm of the $s$-dimensional loss vector over the $s$ servers. Over a sequence of $T$ days, there is a decision maker communicating with the servers which each day has to play an expert with the goal of minimizing the regret compared to the best expert in hindsight. With unlimited communication between the servers, the decision maker can learn the full loss vector and use a standard regret minimizing algorithm, so the interesting question that the paper considers is tradeoffs between regret and communication. Improving upon previous work, they show that (ignoring polylog factors), there is an  algorithm which achieves (normalized) regret $R$ using $O(n/R^2+s/R^2)\max(s^{1-2/p},1)$ bits of communication.

The techniques used in the paper are based on sampling. Instead of all servers communicating the partial loss of an expert, they scale the losses by ($1/p$ powers of) iid exponential random variables and only send the loss of an expert if it exceeds some threshold. The idea is that the maximum of such scaled variables can be turned into an unbiased estimator for the $\ell_p$-norm and additionally, by the thresholding hopefully only a small number of servers will actually have to send an input.
\paragraph{Strengths}
I found the paper to be quite well motivated, since the problem feels motivated from both a theoretical and practical perspective. I think that the algorithmic ideas in the paper are nice and appear to be a natural approach for the problem.

**Strengths:**

I found the paper to be quite well motivated, since the problem feels motivated from both a theoretical and practical perspective. I think that the algorithmic ideas in the paper are nice and appear to be a natural approach for the problem.

**Weaknesses:**

My main concern about the paper is that there are some important steps in the proof where I have doubts about the correctness of the argument.

(1) This is probably my most important concern and is about the important lemma 3.3. The proof is quite sketchy. It is stated that a conditional expectation (l755) equality holds for all realizations of $p_t$. Realizations of what? I assume of $p(t)$, but what is $p(t)$? I assume the probability vector of playing each expert at time $t$ because then the equation in l755 seems to make sense. However, on the next page, it is shown that $\textbf{E}[C_t \mid p_t]=\textbf{E}[C'_t \mid p_t]$ and then concluded that  $\textbf{E}[C_t]=\textbf{E}[C'_t]$. The conclusion seems wrong, because the $p_t$ are realizations of different random variables in the first equation, namely those obtained from running the algorithm with $\hat s_i(t)$ as opposed to $L_i(t)$. So it seems that the paper has proved something like
$\textbf{E}[C_t \mid X=p_t]=\textbf{E}[C'_t \mid Y=p_t]$
for two different variables $X,Y$, but I don't think this implies that $\textbf{E}[C_t]=\textbf{E}[C'_t]$

(2) You claim in line 305 that the probability that the max is not send the the paper $1-1/poly(nT)$, but I don't see where you prove this. I can only find the places where you prove an upper bound on the number of values sent to the coordinator.  It's possible that I just missed it, so in that case, you can just point out to me where you prove it.

These concerns are the reason for my low score. It is however possible that I'm missing something, so if the authors can satisfactorily address the above concerns, I would be very happy to update the score.

**Questions:**

Please address the two concerns under weaknesses.

A couple of more minor comments:

Abstract: Do you mean $R\lesssim$?

l105: $[1,5]$ is $[a,b]$ in the theorem

Theorem 2.1: The bound should be $\exp(-\mu\delta^2/3)$

Is the max stability property your own result?

l327: Shared randomness seems like an unstated model assumption. You should probably discuss this earlier.

Algorithm 3: What does it play with probability $1-\rho$ when it continues to the next time?
\newpage

\section{Nearly Space-Optimal Graph and Hypergraph Sparsification in Insertion-Only Data Streams}

---

> ### Author Response · Authors · 2025-11-21
>
> > These concerns are the reason for my low score. It is however possible that I'm missing something, so if the authors can satisfactorily address the above concerns, I would be very happy to update the score.
>
> > (1) This is probably my most important concern and is about the important lemma 3.3. The proof is quite sketchy. It is stated that a conditional expectation (l755) equality holds for all realizations of $p_t$. Realizations of what? I assume of $p(t)$, but what is $p(t)$? I assume the probability vector of playing each expert at time $t$ because then the equation in l755 seems to make sense. However, on the next page, it is shown that $\mathbf{E}[C_t|p_t]=\mathbf{E}[C'_t|p_t]$ and then concluded that $\mathbf{E}[C_t]=\mathbf{E}[C'_t]$. The conclusion seems wrong, because the $p_t$ are realizations of different random variables in the first equation, namely those obtained from running the algorithm with $\widehat{s}_i(t)$ as opposed to $L_i(t)$. So it seems that the paper has proved something like $\mathbf{E}[C_t|X=p_t]=\mathbf{E}[C'_t|Y=p_t]$ for two different variables $X,Y$, but I don't think this implies that $\mathbf{E}[C_t]=\mathbf{E}[C'_t]$
>
> We apologize for the confusion. Let us clarify. $p_t$ is the probability distribution over experts at time $t$, which is a random variable that depends on the history of losses up to time $t-1$. The losses $\widehat{s^i(t)}$ are also random variables, independent of the probability distribution $p_t$ chosen by MWU. $C_t$ is the loss of the algorithm at time $t$ when using the estimated losses $\widehat{s^i(t)}$, and $C_t'$ is the loss of the algorithm at time $t$ when using the true losses $L_i(t)$. We have $\mathbf{E}[C_t|p_t]=\sum_{i\in[n]} p_i(t)\cdot\mathbf{E}[\widehat{s^i(t)}|p_t]$. Since $\widehat{s^i(t)}$ is an unbiased estimator of $L_i(t)$ independent of $p_t$, we have $\mathbf{E}[\widehat{s^i(t)}|p_t]=L_i(t)$. Thus, $\mathbf{E}[C_t|p_t]=\sum_{i\in[n]} p_i(t)L_i(t)=\mathbf{E}[C'_t|p_t]$, which is the expected loss at time $t$ using the true losses $L_i(t)$ conditioning on the same probability distribution $p_t$. Thus, $\mathbf{E}[C_t]=\mathbf{E}[C_t|p_t]=\mathbf{E}[C'_t|p_t]$. We have clarified this in the updated version of the paper.
>
> We thank the reviewer for pointing this out. We agree that the original proof of Lemma 3.3 was too sketchy and have now replaced it with a more direct argument from first principles. In the revised version, we analyze MWU when it receives unbiased noisy loss estimates $\widehat{s_i(t)}$ with bounded second moment. Specifically, we assume $\mathbf{E}[\widehat{s_i(t)} \mid p_t] = L_i(t)$ and $\mathbf{E}[\widehat{s_i(t)}^2 \mid p_t] \le \rho$, where $p_t$ is the probability vector over experts chosen by MWU at round $t$.
>
> The new proof follows the standard MWU potential analysis but adapts two key steps:
> 1. We take conditional expectations with respect to $p_t$ to handle the randomness of $\widehat{s_i(t)}$ instead of working with deterministic losses.
> 2. We replace the deterministic inequality $e^{-\eta L_i(t)} \le 1 - \eta L_i(t) + \eta^2 L_i(t)^2$ by its conditional expectation $\mathbf{E}[e^{-\eta \widehat{s_i(t)}} \mid p_t] \le 1 - \eta L_i(t) + \eta^2 \rho$.
>
> These are the only differences from the standard MWU analysis, and the remainder of the argument proceeds identically: we bound the expected change in the total weight, apply Jensen’s inequality, and optimize $\eta$ to obtain the expected regret $O(\sqrt{\rho \log n / T})$.
>
> The argument is conceptually straightforward and likely well-known, though we were unable to find an explicit reference for this exact formulation. We have added the full proof to the appendix for clarity.
>
> > (2) You claim in line 305 that the probability that the max is not send the the paper $1-1/poly(nT)$, but I don't see where you prove this. I can only find the places where you prove an upper bound on the number of values sent to the coordinator. It's possible that I just missed it, so in that case, you can just point out to me where you prove it.
>
> We apologize for the confusion. The probability that the maximum is not sent is the probability that $\max_{j\in[s]} q_i^{(b)}(j,t)<100\log(nsT)\cdot s^{1/p}$. By Lemma 2.3 and Lemma 3.2, $\max_{j\in[s]} q_i^{(b)}(j,t)$ is distributioned as $C_{3.2}\cdot E^{1/p}\cdot L_i(t)$, where $E$ is an exponential random variable with rate $1$. Thus, the probability that the maximum is not sent is at most the probability that $E>\left(\frac{C_{3.2}\cdot s^{1/p}}{100 L_i(t) \log(nsT)}\right)^p$. From the probability density function of exponential random variables, this probability is at most $\exp\left(-\frac{(100 \log(nsT))^p}{C_{3.2}^p}\right)\le\frac{1}{\text{poly}(nsT)}$, as desired. We have added this argument to the appendix.

---

> ### Author Response · Authors · 2025-11-21
>
> > A couple of more minor comments:
>
> > Abstract: Do you mean $R\preceq$?
>
> No, our abstract is correct as stated. Our algorithm can only handle regret larger than roughly $O\left(\frac{1}{\sqrt{T}}\right)$. In fact, all possible algorithms must achieve regret at least $O\left(\frac{1}{\sqrt{T}}\right)$, as this is the information-theoretic limit, e.g., see the discussion in Section 1 on optimal regret. Thus, our algorithm can achieve the full range of possible regret parameters.
>
> > l105: $[1,5]$ is $[a,b]$ in the theorem
>
> Thanks for catching this, we have removed this sentence.
>
> > Theorem 2.1: The bound should be $-\exp(-\mu\delta^2/3)$
>
> Yes, thank you for pointing this out. We have updated the manuscript.
>
> > Is the max stability property your own result?
>
> The max stability property of exponential random variables is a known result. We provide a proof in the appendix for completeness.
>
> > l327: Shared randomness seems like an unstated model assumption. You should probably discuss this earlier.
>
> We have mentioned this in the preliminaries (line 85) of the initial submission that all servers share public randomness.
>
> > Algorithm 3: What does it play with probability $1-\rho$ when it continues to the next time?
>
> In this case, the algorithm does not update the weights of the experts and plays the same distribution as in the previous round. We have clarified this in the updated version of the paper.
>
> > \section{Nearly Space-Optimal Graph and Hypergraph Sparsification in Insertion-Only Data Streams}
>
> We are not sure what this refers to, as it does not seem there are any such sections/text in our initial submission. Could you please elaborate what you mean?

---

### Official Review · Reviewer_6CRv · 2025-11-01

**Soundness:** 3
**Presentation:** 4
**Contribution:** 2
**Rating:** 6
**Confidence:** 3

**Summary:**

This paper studies the distributed experts problem in the message-passing model, where the losses for each expert are distributed across multiple servers. The authors introduce a novel distributed protocol that achieves improved communication-regret trade-offs for general $\ell_p$ losses. Specifically, for any $R \geq 1/\sqrt{T}$, they design an algorithm attaining regret $\mathcal{O}(R s^{1/p} \sqrt{\log n})$
using
$
\mathcal{O}\left( \frac{n+s}{R^2} \cdot \max(s^{1-2/p}, 1) \, \mathrm{polylog}(nsT) \right)
$
bits of communication. This work extends beyond prior results, which were restricted to $\ell_1$ bounded losses.

**Strengths:**

-  This is the first work to analyze distributed experts in the coordinator model for general $\ell_p$ losses.

- The embedding of $\ell_p$ losses into $\ell_\infty$ through exponential random variables, combined with a geometric mean estimator for variance reduction, represents a technically sophisticated and creative approach.

- The presentation of three successive algorithms (Algorithms 2--4), a warm-up, a parameterized version that achieves regret-communication tradeoff, and the final algorithm, which does not require the losses to be bounded, offers a clear conceptual progression

- The proofs are rigorous, clearly presented, and appear technically correct.

**Weaknesses:**

- The authors state that it is information-theoretically impossible to achieve regret smaller than $O(1/\sqrt{T})$, but then compare their method to the algorithm of [1] for $R = O(1)$, claiming improved communication in that regime. This causes confusion as $R = O(1)$ is not achievable. Instead, taking $R \ge 1/\sqrt{T}$ seems to reproduce the communication cost of [1], showing no improvement compared to [1]. The claimed improvement in communication complexity should be either removed or clarified.

-  It is unclear why the $\ell_p$ loss is preferable to the additive $\ell_1$ loss, which naturally decomposes across servers and simplifies communication. The paper would benefit from a stronger motivation for why handling these types of losses is relevant.

- The paper lacks formal communication lower bounds, making it unclear whether the proposed protocols are optimal or merely improve upon previous works.

[1]  Zhihao Jia, Qi Pang, Trung Tran, David Woodruff, Zhihao Zhang, and Wenting Zheng. Communication bounds for the distributed experts problem. arXiv preprint 2025

**Questions:**

- The paper states that it is information-theoretically impossible to achieve regret smaller than $O(1/\sqrt{T})$, yet later compares its communication performance to [1] at $R = O(1)$. Could the authors clarify this apparent inconsistency and explain how their claimed improvement holds for the achievable regime $R \ge 1/\sqrt{T}$?

- What are the practical or theoretical motivations for studying $\ell_p$ losses with $p > 1$ in this setting? Since the $\ell_1$ loss decomposes naturally across servers, it would be helpful to better understand scenarios where $\ell_p$ losses provide clear advantages.

 - The paper does not provide matching lower bounds on communication complexity. Can the authors comment on whether such bounds are known, or how close their algorithms might be to optimal in this regard?

[1]  Zhihao Jia, Qi Pang, Trung Tran, David Woodruff, Zhihao Zhang, and Wenting Zheng. Communication bounds for the distributed experts problem. arXiv preprint 2025

---

> ### Author Response · Authors · 2025-11-21
>
> > The authors state that it is information-theoretically impossible to achieve regret smaller than $O(1/\sqrt{T})$, but then compare their method to the algorithm of [1] for $R=O(1)$, claiming improved communication in that regime. This causes confusion as $R=O(1)$ is not achievable. Instead, taking $R\ge 1/\sqrt{T}$ seems to reproduce the communication cost of [1], showing no improvement compared to [1]. The claimed improvement in communication complexity should be either removed or clarified.
>
> > [1] Zhihao Jia, Qi Pang, Trung Tran, David Woodruff, Zhihao Zhang, and Wenting Zheng. Communication bounds for the distributed experts problem. arXiv preprint 2025
>
> We think this confusion might be notational. It is not possible to achieve regret smaller than $\frac{C}{\sqrt{T}}$ for some constant $C>0$. It is possible to achieve regret $R=O(1)$ that is larger than $\frac{C}{\sqrt{T}}$. In particular, when $R=\Theta(1)$, then our protocol achieves regret $R$ using communication roughly $O(s)$, while the protocol of [1] achieves regret $R$ using communication $O(Ts)$. We have clarified this by changing the regret parameterization from $R=O(1)$ to $R=\Theta(1)$.
>
> > It is unclear why the $\ell_p$ loss is preferable to the additive $\ell_1$ loss, which naturally decomposes across servers and simplifies communication. The paper would benefit from a stronger motivation for why handling these types of losses is relevant.
>
> We agree that the $\ell_1$ loss can be natural in some distributed settings.  However, there are many applications where other norms are more suitable. For example, in risk-sensitive optimization, we may want to minimize the maximum loss across servers, which corresponds to the $\ell_\infty$ norm. In other applications, we may want to penalize large deviations, which corresponds to the $\ell_2$ norm. We discuss these motivations in the introduction.
>
> > The paper lacks formal communication lower bounds, making it unclear whether the proposed protocols are optimal or merely improve upon previous works.
>
> We agree that communication lower bounds are an interesting direction for future work. However, we believe that our upper bounds are already significant, as they improve upon previous work and are the first results for general $\ell_p$ losses in the coordinator model.
>
> > The paper states that it is information-theoretically impossible to achieve regret smaller than $O(1/sqrt{T})$, yet later compares its communication performance to [1] at $R=O(1)$. Could the authors clarify this apparent inconsistency and explain how their claimed improvement holds for the achievable regime $R\ge 1/\sqrt{T}$?
>
> > What are the practical or theoretical motivations for studying $\ell_p$ losses with $p>1$ in this setting? Since the $\ell_1$ loss decomposes naturally across servers, it would be helpful to better understand scenarios where $\ell_p$ losses provide clear advantages.
>
> > The paper does not provide matching lower bounds on communication complexity. Can the authors comment on whether such bounds are known, or how close their algorithms might be to optimal in this regard?
>
> Please see our above responses.

---

### Official Review · Reviewer_xgxY · 2025-11-03

**Soundness:** 3
**Presentation:** 4
**Contribution:** 3
**Rating:** 8
**Confidence:** 4

**Summary:**

The paper considers the distributed best-expert problem where there are n experts and s servers. Losses are a matrix L in R^{n x s}, where each column sits on a server, and the Lp norm of a row corresponds to an expert's loss. The goal is to design a distributed online protocol that minimizes regret with respect to the best fixed expert in hindsight, and experiences bounded communication costs per-round.

They make use of the following facts. 1) Taking the maximum across a row of L, where each component is divided by an appropriately scaled exponential random variable, allows one to construct an unbiased estimate of the Lp norm of the row. 2) Since this is a maximum, communication costs can be preserved by omitting components that do not a meet a threshold while still exactly computing (1), so long as all components are not thresholded. Moreover, there is a threshold that ensures a polylog number of samples are sent for each expert, and a vanishingly small probability of failure. 3) By repeating this experiment O(1/p) times, one can reduce the variance of this estimate at the cost of a small amount of bias.

Putting these facts together allows one to execute the multiplicative weights algorithm on a centralized server that receives n*polylog(...) samples per-step via the procedure above, and an addition O(s) communication to check that all servers are done communicating.

Next, they parameterize this algorithm to achieve a trade-off between regret and communication in a very trivial way. With some probability, the servers can be made to skip the protocol all-together. By sharing randomness, this brings communication complexity down to zero on those rounds, while maximizing regret, and increasing the variance of the loss estimates. For a target regret bound R, they show this protocol requires (n + s)(polylog(...))/R^2 communication complexity. Finally, they employ one more trick, where the threshold selected in step (2) is allowed to be scaled down according to an appropriately chosen distribution to improve the dependency on s even further.

Empirical evaluations are given on a hyperparameter tuning task, showing improved communication costs  over Jia et al.

**Strengths:**

Strength #1: The paper is extremely well written, communicating complicated ideas in an incremental fashion. The main ideas of the theory are presented in a manner that allow the proofs to be checked easily.

Strength #2: The paper improves upon previous work by a) generalizing to arbitrary p-norms, b) attaining the regret-sensitive rate s/R^2 on the dependency on the number of servers, and c) giving an algorithm that improves the dependence on number of servers by a factor max(s^{1-2/p})

Strength #3: I found the approach in the warm-up algorithm to be very elegant, and did a good job of setting up the more technical details of the later algorithms.

**Weaknesses:**

Weakness #1: The trick to trade-off communication with regret is somewhat artificial, requiring that all servers are silent on some rounds. While the theory works out, I wonder if there is a more natural algorithm that would yield further improvement.

Weakness #2: This is largely a theory paper exploring communication-regret trade-offs in expert learning. It may be a little outside the primary areas of interest for ICLR.

**Questions:**

Fig 1 seems like it is underselling the improvement for small p. For p = 1, does this work attain communication complexity, (n /(s R^2)) polylog(...) through theorem 1.3? Or am I mistaken?

The prose in Section 5 claims that losses need not be bounded, and yet the Lemmas assume l_i(j, t) <= 1. Can the authors explain?

Do the authors think there might exist a more natural protocol that avoids having all servers go silent on some rounds? While the theory works out, this seems artificial.

"We remark that the choices of the losses being within the range [1, 5] in the statement of Theorem 1.1 are arbitrary" => I think at some point the manuscript was updated to actually state Theorem 1.1 with arbitrary interval [a,b].

There are minor typos in the appendix involving misaligned hat symbols. See bottom of page 16, top of page 17. Please proof-read and fix.

---

> ### Author Response · Authors · 2025-11-21
>
> > Weakness #1: The trick to trade-off communication with regret is somewhat artificial, requiring that all servers are silent on some rounds. While the theory works out, I wonder if there is a more natural algorithm that would yield further improvement.
>
> Yes, our bounds achieving communication sublinear in the total time horizon can only be possible when all servers are silent on some rounds. In particular, if some server is required to speak on each round, then the total communication is at least $\Omega(T)$, which is worse than our results in some regimes of the desired regret $R$. We agree that the identification of other aspects that can be parameterized to yield further improvement is an interesting direction worthy of future study.
>
> > Weakness #2: This is largely a theory paper exploring communication-regret trade-offs in expert learning. It may be a little outside the primary areas of interest for ICLR.
>
> Thanks for the concern. We remark that this work studies learning with experts, a fundamental problem at the intersection of reinforcement learning and learning theory, both of which are explicitly listed in ICLR’s “non-exhaustive list of relevant topics” in the call for papers. Moreover, our work builds upon the results in the recent paper by Jia et al., which appeared in NeurIPS 2024, perhaps indicating that the community has interest in this area of research.
>
> > Fig 1 seems like it is underselling the improvement for small p. For p = 1, does this work attain communication complexity, (n /(s R^2)) polylog(...) through theorem 1.3? Or am I mistaken?
>
> Since $s^{1-2/p}\le 1$ for $p=1$, then the communication complexity of $\left(\frac{n+s}{R^2}\right)\cdot\max(s^{1-2/p},1)\cdot\text{polylog}(nsT)$ stated in Theorem 1.3 corresponds to $\left(\frac{n+s}{R^2}\right)\cdot\text{polylog}(nsT)$ for $p=1$, which matches the bounds claimed in Figure 1.
>
> > The prose in Section 5 claims that losses need not be bounded, and yet the Lemmas assume l_i(j, t) <= 1. Can the authors explain?
>
> Currently, Algorithm 4 has explicit parameters computed for $L_i(j,t)\le 1$. The parameters can be easily adjusted to handle general losses, though the presentation is slightly messier, so the current technical statements make the simplifying assumption that the losses normalized to at most $1$. In general, if the losses are between an arbitrary interval $[a,b]$, then the natural approach of changing the thresholds in Algorithm 4 by a factor of $b$, as well as the technical statements by a factor of $b$, is enough to handle general losses.
>
> > Do the authors think there might exist a more natural protocol that avoids having all servers go silent on some rounds? While the theory works out, this seems artificial.
>
> We agree that there may be other algorithms that yield further improvement. For example, rather than having all servers be silent, another possible algorithm would be to subsample the servers in each round. However, it seems challenging to analyze the resulting regret because the $\ell_p$ norm does not naturally decompose into a sum of terms, unlike the $\ell_1$ norm. This is why we embed the $\ell_p$ norm into the $\ell_\infty$ norm using exponential random variables. We agree that other approaches, such as subsampling the servers, are interesting directions for future work.
>
> > "We remark that the choices of the losses being within the range [1, 5] in the statement of Theorem 1.1 are arbitrary" => I think at some point the manuscript was updated to actually state Theorem 1.1 with arbitrary interval [a,b].
>
> Thanks, we have removed this statement and the following line about the losses being within a constant factor.
>
> > There are minor typos in the appendix involving misaligned hat symbols. See bottom of page 16, top of page 17. Please proof-read and fix.
>
> Thanks for the pointer, we have fixed these typos.

---

### Author Response · Authors · 2025-11-21

We thank the reviewers for their thoughtful and detailed comments. We also appreciate the positive feedback on our paper, including:
- The paper is extremely well written, communicating complicated ideas in an incremental fashion. (Reviewer xgxY)
- The main ideas of the theory are presented in a manner that allow the proofs to be checked easily. (Reviewer xgxY)
- The paper improves upon previous work by a) generalizing to arbitrary p-norms, b) attaining the regret-sensitive rate s/R^2 on the dependency on the number of servers, and c) giving an algorithm that improves the dependence on number of servers by a factor max(s^{1-2/p}) (Reviewer xgxY)
- I found the approach in the warm-up algorithm to be very elegant, and did a good job of setting up the more technical details of the later algorithms. (Reviewer xgxY)
- This is the first work to analyze distributed experts in the coordinator model for general $\ell_p$ losses. (Reviewer 6CRv)
- The embedding of $\ell_p$ losses into $\ell_\infty$ through exponential random variables, combined with a geometric mean estimator for variance reduction, represents a technically sophisticated and creative approach. (Reviewer 6CRv)
- The presentation of three successive algorithms (Algorithms 2--4), a warm-up, a parameterized version that achieves regret-communication tradeoff, and the final algorithm, which does not require the losses to be bounded, offers a clear conceptual progression (Reviewer 6CRv)
- The proofs are rigorous, clearly presented, and appear technically correct. (Reviewer 6CRv)
- I found the paper to be quite well motivated, since the problem feels motivated from both a theoretical and practical perspective. (Reviewer zz4T)
- I think that the algorithmic ideas in the paper are nice and appear to be a natural approach for the problem. (Reviewer zz4T)
- The paper is well written (Reviewer g7FC)
- The analysis appears to be sound (Reviewer g7FC)

We have uploaded a revised version of the manuscript in which we addressed the initial reviewer comments and clarified several technical points. In particular, we replaced the proof sketch of Lemma 3.3 with a fully detailed argument using conditional expectations and bounded second moments, added the missing probability calculation for sending the maximum to the coordinator, corrected minor typos throughout the appendix, and updated statements regarding loss intervals to arbitrary $[a,b]$. We believe these revisions substantially improve the clarity and completeness of the manuscript.

We believe these changes have helped improve the presentation and clarity of the manuscript and have addressed the initial reviewer concerns. We respond in more detail to individual points below and welcome further discussion; please let us know if you feel any concerns have not been fully resolved.

---

### Meta-Review · Area_Chair_rCkB · 2026-01-06

**Summary:**

This paper analyzes the regret-communication tradeoff in distributed expert problem. It improves previous theoretical results by achieving near-optimal regret for general l_p losses while keeping communication provably low.

**Reviewer Concerns:**

Two reviewers are positive about the theoretical contributions of this work, while the other two reviewers had some concerns. One reviewer had some questions regarding the technical proof details. The other reviewer questioned about the technical novelty of this work, as the techniques were used in other existing works.

During the rebuttal, the authors provided detailed responses to the technical questions, which seem reasonable and convincing. Regarding the technical novelty, as the authors stated in their paper, "although similar approaches have been used in other contexts such as norm estimation in the streaming model (Li, 2008; Woodruff & Zhou, 2021), this is the first time these techniques have been applied to online learning in the distributed setting". Based on my own reading, I find the results to be non-trivial and the contributions to be solid.

**Reviewer Scores:**

The scores were quite diverse: 8/6/2/2. One reviewer with score 2 would likely increase their score to be positive, as they had some technical questions, which are addressed by the authors during rebuttal.

---

### Decision · Program_Chairs · 2026-01-26

Accept (Poster)